# Inner Speech as Behavior Guides: Steerable Imitation of Diverse Behaviors for Human-AI coordination

**Rakshit S. Trivedi**[*†]
Massachusetts Institute of Technology
triver@mit.edu

**Kartik Sharma**[*]
Georgia Institute of Technology
ksartik@gatech.edu

**David C. Parkes**
Harvard University
parkes@eecs.harvard.edu

## Abstract

Effective human-AI coordination requires artificial agents capable of exhibiting and responding to human-like behaviors while adapting to changing contexts. Imitation learning has emerged as one of the prominent approaches to build such agents by training them to mimic human-demonstrated behaviors. However, current methods struggle to capture the inherent diversity and non-Markovian nature of human behavior and lack the ability to steer behavior at inference time. Drawing inspiration from the theory of human cognitive processes, where inner speech guides action selection before execution, we propose *MIMIC (Modeling Inner Motivations for Imitation and Control)*, a framework that uses language as an internal representation of behavioral intent. MIMIC employs the novel use of vision-language models as linguistic scaffolding to train a conditional variational autoencoder capable of generating inner speech from observations. A diffusion-based behavior cloning policy then selects actions conditioned on current observations and the generated inner speech. MIMIC enables fine-grained steering of behavior at inference time by conditioning the agent on behavior-specific speech. Experiments across robotic manipulation tasks and human-AI collaboration games demonstrate that MIMIC significantly enhances both behavior diversity and fidelity to human demonstrations while enabling nuanced behavioral steering without training on additional demonstrations. We open source our code and provide pre-trained MIMIC agents and qualitative demos at: https://mimic-research.github.io.

## 1   Introduction

Human-AI collaboration in complex settings requires artificial agents that can anticipate, understand, and appropriately respond to the full spectrum of human behavior. This capability appears important in ensuring AI safety and alignment with human values and expectations [5, 7, 28]. One direction towards progress is to develop artificial agents which are able to mimic human behavioral patterns. Through *in silico* surrogates for the richness of human behavior, we can hope to support the safe deployment of AI technologies involving human-AI collaboration—enabling comprehensive pre-deployment testing and validation across diverse interaction scenarios that would otherwise be impractical or unsafe to assess through direct human participation.

---

[*]Equal contribution
[†]Corresponding author

39th Conference on Neural Information Processing Systems (NeurIPS 2025).

*Imitation learning (IL)* presents a promising paradigm for developing these human-like agents by enabling behavioral acquisition directly from demonstrations. However, the effective design of such agents imposes several requirements: (1) capture multimodal distributions of human behaviors reflecting diverse motivations and skill levels; (2) generate contextually appropriate novel behaviors beyond those demonstrated; (3) provide mechanisms for controlled behavior generation; (4) process visual inputs characteristic of realistic environments; and (5) operate without requiring environment interactions during training.

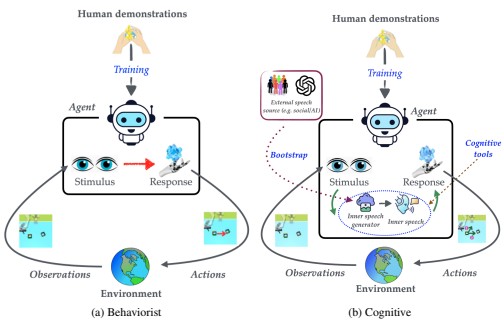

Figure 1: **Paradigm comparison. (a)** direct state-to-action mapping. **(b)** inner speech $m_t$ mediates between perception and action. Extended discussion available in Appendix B.1.

In this work, we focus on *behavior cloning (BC)* [34], which uses supervised learning to model human demonstrations. Despite its theoretical limitations [38], BC offers a simple, efficient, and offline approach to IL that has demonstrated remarkable efficacy across different domains [33]. Recent advances in BC extend beyond standard implementations to explicitly address behavioral diversity through the use of transformer models [39, 24] and diffusion-based behavior policies [33, 36]. While achieving state-of-art results, these approaches still leave significant room for improvement with respect to distributional realism. Further, these approaches exhibit fundamental constraints in their capabilities: some lack support for controlled generation entirely [39, 33], while others restrict themselves to goal-conditional generation [24, 36]. This limitation stands in contrast to our objective—enabling steerable imitation and generation of novel behaviors through designer-specified control at inference time—a more general paradigm that subsumes goal-conditional generation while offering significant flexibility in behavior synthesis.

A typical IL model assumes that the human takes her decision given the state as $s_t \mapsto_{\mathcal{H}} a_t$ and aims to directly approximate this conditional probability distribution $\pi_\theta(a \mid s) \approx p_{\mathcal{H}}(a \mid s)$. However, this state-to-action mapping overlooks a crucial insight from psychological and cognitive science literature [44, 42, 6]: human decisions are influenced by intrinsic motivations and *inner speech* that mediate between perception and action, even when not explicitly tied to task objectives. Cognitive science research on inner speech [44, 42], conceptualizes inner speech as an internalized form of language that serves as a *mediational mechanism*[1] between environmental perception and action selection. This inner speech provides a cognitive framework that explains how identical environmental stimuli can produce diverse behavioral responses across different individuals. Figure 1 illustrates this paradigm shift: whereas traditional IL treats behavior as direct state-to-action mapping ($s_t \rightarrow a_t$), the cognitive approach introduces inner speech as a mediational layer ($s_t \rightarrow m_t \rightarrow a_t$), enabling behavioral diversity through internal deliberation. Based on this theoretical foundation, we posit that effective imitation learning should model both the conditional action policy $p_{\mathcal{H}}(a \mid s, m)$ and the inner speech generation process $p_{\mathcal{H}}(m \mid s)$ to better capture how humans act in a given scenario:

**Proposition 1** (Imitating with Inner Speech $m$). *Instead of directly learning the human action distribution conditioned on the environment's state, $\pi_\theta(a \mid s) \approx p_{\mathcal{H}}(a \mid s)$, we propose to model human behavior through inner speech mediation: $P_{\mathcal{H}}(a \mid s) = \int p_{\mathcal{H}}(a \mid s, m) p_{\mathcal{H}}(m \mid s) dm$. Here, inner speech $m$ is: (1) an internal representation that mediates task performance while potentially encompassing broader motivations, (2) represented in natural language, and (3) internal to the agent and not directly observable from demonstrations.*

To this end, we formalize "inner speech as behavior guides", a computational framework building on Vygotsky's mediational theory [44] and Sokolov's empirical observations [42]. We then introduce *MIMIC (Modeling Inner Motivations for Imitation and Control)*[2], a novel imitation framework that operationalizes this theoretical foundation through three key components: (1) an inner speech

---

[1]A mediational mechanism is a cognitive process that transforms environmental stimuli into behavioral responses through an intermediate representation. In this framework, inner speech serves as the intermediate layer that interprets observations before determining actions.

[2]We note that MIMIC draws computational inspiration from cognitive theory rather than claiming biological plausibility. Similar to how convolutional networks leverage principles of visual processing without replicating neural mechanisms, our framework operationalizes functional properties of inner speech through computational architectures without asserting neurobiological correspondence.

conditioned behavior cloner implemented using conditional diffusion-based policy; (2) a behavior guided inner speech generator utilizing a conditional variational autoencoder (CVAE) that captures the stochastic and semantically condensed nature of inner speech; and (3) a vision-language model serving as linguistic scaffolding that provides external language input to train the inner speech generator. During simulation, the agent generates its own inner speech based on ongoing behavior and uses this to condition its policy. MIMIC enables designer control through two mechanisms: (i) accepting textual descriptions of desired behavior as initial inner speech, with the agent continuing to generate its own inner speech as needed, and (ii) allowing designers to set a polling window that determines inner speech generation frequency, thereby serving the dual purpose of finetuning the amount of inner speech and facilitating behavior corrections at regular intervals—a feature particularly valuable in addressing behavior cloning's brittleness.

We evaluate MIMIC's efficacy across three dimensions: (i) fidelity in imitating diverse behaviors, (ii) capability for steerable behavior generation during simulation, and (iii) performance enhancement in human-AI collaborative contexts. For the first two dimensions, we conduct experiments on the *D3IL benchmark dataset* [21], which encompasses diverse robotic manipulation tasks. Our results demonstrate that MIMIC surpasses state-of-the-art BC approaches in generating human-like behaviors, achieving higher entropy, higher fidelity and often higher success rates in its generated trajectories while enabling designer-specified control over behavior generation. Next, we consider the *Overcooked* environment in a 2-player setting [5], where MIMIC agents evaluated in collaboration with human proxy models achieve consistently higher cooperative rewards than agents trained using other BC approaches. We substantiate these empirical findings with comprehensive qualitative analysis and architectural ablation studies. Our results highlight the potential of this inner speech based approach to create adaptive, human-like agents that can act as effective human surrogates, facilitating the safe development and evaluation of AI agents before their deployment among humans.

## 2  Preliminaries and Related Work

**Problem Setup.** Let $(\mathcal{S}, \mathcal{A}, \mathcal{P}, r, \nu, \gamma)$ define a *Markov Decision Process* (MDP), where $\mathcal{S}$ and $\mathcal{A}$ represent state and action spaces, $\mathcal{P} : \mathcal{S} \times \mathcal{A} \times \mathcal{S} \to \mathbb{R}$ is the transition probability distribution, $r$ denotes the reward function, $\nu : \mathcal{S} \to \mathbb{R}$ is the initial state distribution, and $\gamma \in (0, 1)$ is the discount factor. A stochastic behavior policy $\pi : \mathcal{S} \times \mathcal{A} \to [0, 1]$ defines an agent's action selection.

*Imitation learning* focuses on learning policy $\pi$ directly from demonstrations without access to reward signals. We consider expert policy $\pi_E$ as a mixture of experts $\{\pi_E^0, \pi_E^1, ...\}$, representing diverse tasks, state-space specializations, or behavioral variations. A demonstration trajectory $\tau^i = \{(s_0^i, a_0^i), ..., (s_T^i, a_T^i)\}$ records a sequence of state-action pairs. Given dataset $\mathcal{D} = \{\tau^i\}_{i=1}^N$ of $N$ trajectories, **behavior cloning (BC)** applies supervised learning over state-action pairs, maximizing action likelihood: $\mathcal{L}_{BC} = \max_\theta \sum_{i=1}^N \sum_{t=1}^{|\tau_i|} \log \pi_\theta(a_t^i | s_t^i)$. Our technical approach employs diffusion models and conditional variational autoencoders (CVAEs) and refer interested readers to Appendix D for a background discussion on these models.

**Language-Interfaced Imitation Learning.** Recent advances in language-based interfaces for IL include *Thought cloning (TC)* [18], which directly imitates human thoughts but differs from our method in that it requires access to annotation of actual human thought for each step in the demonstration trajectory. Further, its performance is highly tied with goal (mission) conditioning and degrades by almost 40% in our experiments once the goal (mission) condition is removed. Similarly, external speech has also been used to steer agent behaviors through action re-ranking [30] although with mechanisms that remain external to the agent. Closest prior is [46] that use "intra-agent speech" as semi-supervised captioning that is frozen and used as auxiliary supervision during BC to achieve zero-shot object-level generalization. We instead treat inner speech as a latent mediator that conditions the policy and is generated online, enabling steerable, diverse imitation.

**AI Approaches to Modeling Cognitive Processes.** Recent AI research has explored computational implementations of cognitive processes. [6] propose *autotelic AI* which internalizes language for self-directed learning, emphasizing language as a tool for goal generation and intrinsic motivation. While sharing our interest in cognitive foundations, autotelic systems focus on autonomous learning rather than behavioral diversity, using language primarily for goal generation rather than as a mediational mechanism. [19] utilize large language models to simulate reasoning processes (chain of thought) before action selection, implementing serial, deterministic reasoning rather than the stochastic,

parallel processing characteristic of inner speech in Vygotskian theory. Our framework models inner speech as a probabilistic process that generates diverse behavioral patterns, more closely aligning with cognitive theories of human behavioral diversity. [4] propose natural language as a latent space for reinforcement learning, using language to give a hierarchical structure to behavior. While this approach shares our recognition of language as a cognitive tool, it focuses on decomposing complex tasks rather than generating behavioral diversity. Our framework uniquely combines the stochastic nature of inner speech with IL to capture human behavioral variation without explicit linguistic supervision. An extended related work discussion is available in Appendix C.

## 3    MIMIC: **Inner Speech as Behavior Guides**

As discussed in Section 1, existing IL techniques [18, 30, 39, 24, 46, 43] do not satisfy Proposition 1, and struggle to fully capture the *noisy*, *diverse*, and *non-Markovian* nature of human behavior. Here, we first present a theoretical formulation that connects inner speech and behavioral diversity, grounded in Vygotsky's characterization of inner speech as a mediational mechanism. We then present MIMIC, a novel imitation framework that operationalizes this theoretical model to enable agents to generate inner speech and condition their action selection on these representations. Our framework enables steerable generation of diverse behaviors: by conditioning on user-specified speech and generating its own inner speech representations, the agent can adopt corresponding behavioral modes.

### 3.1    Theoretical Formulation of Inner Speech

The theoretical framework conceptualizes inner speech as a mediational mechanism for capturing behavioral diversity in imitation learning.

#### 3.1.1    Inner Speech as a Generative Mediating Process

Drawing from Vygotsky's [44] theoretical characterization of inner speech as a cognitive mediator between perception and action, we formalize inner speech as a stochastic process that transforms environmental observations into linguistic representations before generating behavior. For an agent operating in environment $\mathcal{E}$, we define: $\mathcal{S}$ as the *state space*, representing environmental percepts; $\mathcal{A}$ as the *action space*, encompassing possible behavioral responses; $\mathcal{Z}$ as the *inner speech representation space*, a latent embedding of verbalized cognition; $f_\phi : \mathcal{S} \rightarrow \mathcal{Z}$ as the *inner speech generation function*; and $g_\theta : \mathcal{S} \times \mathcal{Z} \rightarrow \mathcal{A}$ as the *action policy conditioned on observation and inner speech*.

Vygotsky identified following structural properties characterizing inner speech that inform our computational design: **(i) Predicativity**: Inner speech emphasizes relationships and actions rather than entities. For example, an agent internally represents "moving left to coordinate" rather than "the box is on the left". **(ii) Semantic Condensation**: Complex strategic meanings compress into compact representations while preserving functional significance, e.g. a multi-step coordination strategy becomes a brief internal directive. **(iii) Regulatory Dynamics**: As Vygotsky conceptualized inner speech as enabling organization of behavior—regulating responses over time—we formalize this as temporal regulatory dynamics. Inner speech generation operates over extended timescales, conditioned on behavioral history rather than instantaneous state, reflecting that strategic processing integrates information across temporal windows. This is consistent with empirical observations that inner speech intensifies during complex cognitive tasks [42]. These properties indicate that inner speech operates as a compressed, semantically enriched representation—reduced in dimensionality relative to perceptual input, yet preserving task-relevant strategic information and temporal context.

#### 3.1.2    Mathematical Formulation

We formalize inner speech as a stochastic mediational process grounded in Vygotsky's theoretical characterization. The agent's policy decomposes through inner speech as a latent mediator:

$$p(a|s) = \int_{\mathcal{Z}} p(a|s, m)p(m|s)dm \tag{1}$$

where $s \in \mathcal{S}$ is the environmental state, $m \in \mathcal{Z}$ represents the inner speech embedding, $p(m|s)$ generates inner speech from observations, and $p(a|s, m)$ produces actions conditioned on both state and inner speech. This stochastic formulation captures behavioral diversity: different inner speech representations lead to different behavioral modes even when facing identical environmental states.

We now provide mathematical formalizations of the three structural properties characterized above, showing how each property shapes the computational structure of inner speech generation.

**Predicativity: Relational Structure Extraction.** The emphasis on relationships over entities is formalized through a relational encoding that generates inner speech distributions: $p(m_t|\mathcal{H}_t) = f_\phi(\mathcal{H}_t; \psi_{\text{rel}})$, where $\psi_{\text{rel}}$ denotes parameters that prioritize relational features in the behavioral history $\mathcal{H}_t$. This ensures inner speech captures strategic patterns.

**Semantic Condensation: Information-Theoretic Formulation.** The compression of complex meanings into compact representations is formalized through the information bottleneck framework: $\max_m [I(m; \mathcal{H}_t) - \beta \cdot D_{KL}(p(m|\mathcal{H}_t) \| p(m))]$, where $I(m; \mathcal{H}_t)$ measures mutual information between inner speech and behavioral history, $D_{KL}(p(m|\mathcal{H}_t) \| p(m))$ enforces compression toward a simple prior, and $\beta$ controls the trade-off. Higher $\beta$ enforces stronger compression, modeling the progression from elaborate external descriptions toward abbreviated mature inner speech.

**Temporal Regulatory Dynamics.** We formalize this through non-Markovian conditioning on behavioral history: $p(m_t|\mathcal{H}_t)$, where $\mathcal{H}_t = \{s_{t-W:t}, a_{t-W:t-1}\}$, where $\mathcal{H}_t$ represents a window of length $W$ encompassing recent states $s_{t-W:t}$ and actions $a_{t-W:t-1}$. This formulation reflects that strategic processing emerges from accumulated experience rather than instantaneous perception.

### 3.1.3 Architectural Correspondence

We now describe how our computational architecture operationalizes the theoretical formulation presented above. Our framework explicitly implements predicativity, semantic condensation, and temporal regulatory dynamics through dedicated architectural components.

**Transformer Attention as Predicative Processing.** The emphasis on relational structures is implemented through the transformer's multi-head attention mechanism that processes inner speech alongside state and action representations: $\text{Attention}(Q, K, V) = \text{softmax}\left(\frac{QK^T}{\sqrt{d_k}}\right) V$, where $Q, K, V$ are derived from the concatenated representation $[\mathbf{z}_s, \mathbf{z}_m, \mathbf{z}_a, \mathbf{z}_\tau]$ encoding state, inner speech, action history, and temporal information respectively. The cross-attention mechanism naturally focuses on relationships between inner speech $\mathbf{z}_m$ and behavioral context, capturing task-relevant strategic patterns ("how to coordinate") rather than merely encoding object features ("what is present").

**Variational Compression for Semantic Condensation.** The Conditional Variational Autoencoder architecture directly instantiates the information bottleneck formulation. The encoder compresses behavioral history into a latent bottleneck $q_\phi(z|\mathcal{H}_t) = \mathcal{N}(\mu_\phi(\mathcal{H}_t), \sigma_\phi^2(\mathcal{H}_t))$, from which inner speech codes are sampled as $z \sim q_\phi(z|\mathcal{H}_t)$ and decoded via $p_\theta(m|z, \mathcal{H}_t) = \Psi_{\text{dec}}(z, \mathcal{H}_t)$. The variational objective realizes the information-theoretic trade-off: $\mathcal{L}_{\text{IS}} = \mathbb{E}_{q_\phi(z|\mathcal{H}_t)}[\log p_\theta(m|z, \mathcal{H}_t)] - \beta D_{KL}(q_\phi(z|\mathcal{H}_t) \| p(z))$, where the reconstruction term $\mathbb{E}[\log p_\theta(m|z, \mathcal{H}_t)]$ corresponds to maximizing $I(m; \mathcal{H}_t)$ (preserving behavioral relevance), while the KL divergence term enforces compression toward a simple prior $p(z)$. The annealing parameter $\beta$ models the progressive shift from flexible external linguistic structure to compressed autonomous inner speech generation.

**Periodic Generation as Temporal Regulation.** The non-Markovian temporal structure is implemented through periodic inner speech generation at fixed intervals: $m_t = \Psi_{\text{dec}}(\Psi_{\text{enc}}(\mathcal{H}_t), \mathcal{H}_t)$ if $t \mod W = 0$, and $m_t = m_{t-1}$ otherwise. This $W$-step update cycle captures the intermittent nature of strategic processing. The agent generates new inner speech every $W$ steps based on accumulated behavioral history, then conditions its actions on this inner speech until the next update.

## 3.2 Learning Inner Speech and Behavior Generation from Demonstrations

We now describe how MIMIC learns these components from demonstrations. The framework consists of two key elements—an inner speech-conditioned behavior cloner and a behavior-conditioned inner speech generator—implementing the theoretical formulation presented above (c.f. 2).

### 3.2.1 Inner Speech-conditioned Behavior Cloner

Following Proposition 1, we learn the space of human actions in the presence of an inner speech $m$. Given a dataset of demonstrations $\mathcal{D} = \{(s_t^{(i)}, a_t^{(i)}) \mid t \in [1, T], i \in [1, n]\}$, we augment it with initial speech $m^{(i)}$ to obtain $\mathcal{D}_M = \{(m^{(i)}, s_t^{(i)}, a_t^{(i)}) \mid t \in [1, T], i \in [1, n]\}$. We then train an

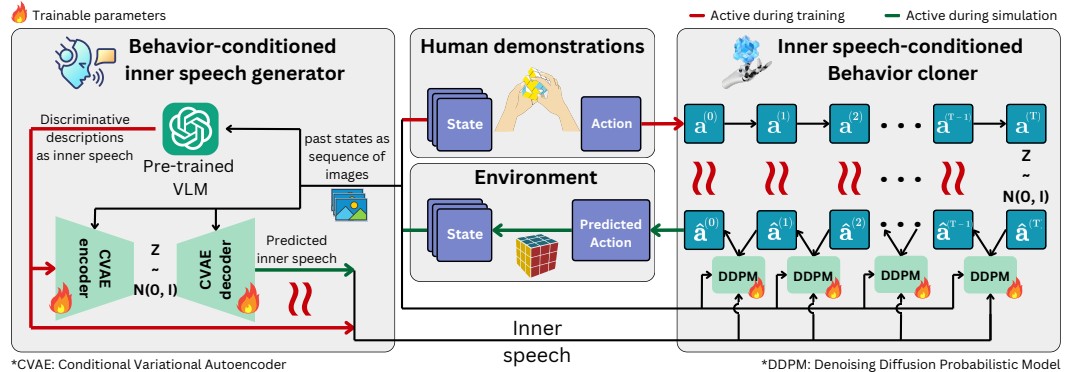

Figure 2: **Overview of MIMIC**: Agent inner speech is scaffolded by using a pre-trained VLM to discriminate different behaviors in human demonstrations. Next, we train a DDPM-T (diffusion policy with transformer architecture) behavior cloner conditioned on this inner speech and a VAE-based inner speech generator conditioned on the history of states. During simulation, the inner speech is periodically generated to influence the behavior cloner given the actions it generated in the past.

imitator $\pi_\theta(a \mid s, m)$ that models the probability of an action given the environment's state and the inner speech $m$ of the agent. We implement the action policy $g_\theta$ in the theoretical framework using a *diffusion-based policy with a transformer architecture (DDPM-T)* [33]. This architecture trains a transformer-based conditional denoising network $\hat{\epsilon}_\theta(\hat{\mathbf{a}}_\tau, s, m, \tau)$ to predict the noise added to the action $\hat{\mathbf{a}}$ at step $\tau$ given the current state $s$ and inner speech $m$. The diffusion process follows:

$$\begin{cases} \textbf{Forward diffusion: } \mathbf{a}_{\tau+1} \sim \mathcal{N}(\sqrt{1-\beta_\tau}\mathbf{a}_\tau, \beta_t\mathbf{I}) \\ \textbf{Reverse diffusion: } \hat{\mathbf{a}}_\tau \sim \mathcal{N}(\frac{1}{\sqrt{1-\beta_\tau}}(\hat{\mathbf{a}}_{\tau+1} - \beta_\tau\hat{\epsilon}_\theta(\hat{\mathbf{a}}_{\tau+1}, s, m, \tau)), \beta_\tau\mathbf{I}) \end{cases} \quad (2)$$

We train this network using a *reconstruction Fisher divergence loss*:

$$\mathcal{L}_{\text{diff}}(\mathcal{D}_M) = \mathbb{E}_{(s,\mathbf{a}_0,m)\sim\mathcal{D}_M, \tau\sim[1,T_D]}\|\hat{\epsilon}(\mathbf{a}_\tau, s, m, \tau) - \epsilon\|_2^2. \quad (3)$$

The network is trained using classifier-free guidance, where m is randomly replaced with $\mathbf{0}$ with probability $p$ during training [16]. This enables the model to operate both with and without inner speech conditioning. $\hat{\epsilon}$ is a transformer architecture that can take inputs of arbitrary size. We first encode the inputs $s, m, \hat{\mathbf{a}}$, and $\tau$ using different encoders and then concatenate the representations $\mathbf{z}_s, \mathbf{z}_m, \mathbf{z}_{\hat{\mathbf{a}}}$, and $\mathbf{z}_\tau$ together before passing into the transformer. The state $s$ is encoded using domain-specific encoders such as a *vision-based convolutional encoder* for a vision environment or a *locomotion-based feature encoder* for a vision-free environment. Inner speech $m$ will be provided as latent representations in the natural language space and use a trainable 2-layer MLP to obtain $\mathbf{z}_m$. $\mathbf{z}_{\hat{\mathbf{a}}}$ and $\mathbf{z}_\tau$ are obtained using standard linear encoding and cosine-based encoding, respectively.

### 3.2.2 Behavior-Conditioned Inner Speech Generator

This component instantiates the inner speech generation function $f_\phi : \mathcal{S} \to \mathcal{Z}$. We implement it using a conditional variational autoencoder (CVAE) architecture to capture the stochastic and semantically condensed nature of inner speech. The variational nature of this implementation models the probabilistic nature of cognitive self-regulation, allowing for the generation of diverse yet contextually appropriate inner speech representations.

**Algorithm 1 (Appendix A): Training and Vision-Language Model Scaffolding.**

The training process for the inner speech generator implements a learning-based internalization of linguistic structure. Central to this process is the use of vision-language models (VLMs) to provide external linguistic scaffolding. The VLMs generate initial descriptive characterizations of demonstrated behaviors, providing explicit linguistic structure that serves as training targets for the CVAE. We model the agent's inner speech $m$ as linguistic descriptions of behavior. For each demonstration in our dataset, we obtain a sequence of T images $(\mathbf{I}_1^{(i)}, \mathbf{I}_2^{(i)}, \cdots, \mathbf{I}_T^{(i)})$, which are converted into a GIF where T is the task horizon. We then use a VLM to generate descriptive external speech that characterizes the behavior by passing k=8 randomly picked GIFs with the prompt:

```
Immerse yourself in the role of the {agent} that has enacted the actions in the
attached GIFs. {environment}. Generate your inner thought process that helps describe
the distinctive behaviors shown in the each of the GIF. ONLY generate those phrases
that differentiate the behavior adopted in different GIFs. YOU MUST return the thought
process of each GIF in a new line.
```

This process generates linguistic descriptions $c^{(i)}$ of each demonstration's behavior, which we encode using the CLIP [35] embedding model to obtain $m^{(i)}$. Figure 3 shows the latent space of inner speech obtained from demonstrations, exemplifying the diversity captured by inner speech. These external descriptions serve as initial scaffolding that the model learns to compress and internalize, implementing the semantic condensation principle.

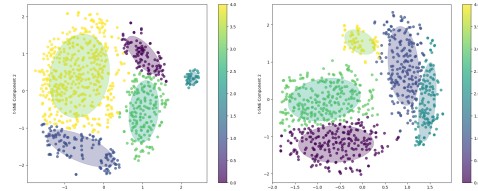

Figure 3: TSNE visualization of inner speech for Aligning dataset. (c.f. Appendix G for others.)

By using pre-trained VLMs, we leverage their rich linguistic knowledge to generate descriptions that capture behaviorally-relevant attributes and strategic patterns. Critically, this scaffolding approach circumvents the need for explicit human annotation of thought during demonstrations, making MIMIC more scalable and broadly applicable than approaches requiring verbalized thoughts at each step [18].

For computational efficiency, we train a specific generative model $\Psi$ to generate realistic inner speech by only looking at a limited historical behavior. We use a conditional variational auto-encoder (CVAE), where both encoder $\Psi_{\text{enc}}$ and decoder $\Psi_{\text{dec}}$ are conditioned on the past image sequence, which is encoded by pooling over the convolutional representations. The CVAE is trained to generate the CLIP-encoded inner speech generated by the oracle VLM by minimizing the loss function:

$$\mathcal{L}_{\text{is}}(\mathcal{D}_M) = \sum_{i=1}^{n} \sum_{t=W}^{T} \left\| m^{(i)} - \Psi_{\text{dec}}(\Psi_{\text{enc}}(m^{(i)}, \mathbf{I}_{t-W:t}^{(i)}), \mathbf{I}_{t-W:t}^{(i)}) \right\|_2^2 + \beta \, \Delta_{KL}(\Psi_{enc}(m^{(i)}), \mathcal{N}(\mathbf{0}, \mathbf{I})), \quad (4)$$

where $\Delta_{KL}$ denotes the re-parameterized KL divergence and $W$ denotes a fixed window size. The regularization parameter $\beta$ is annealed during training as described in the architectural correspondence, modeling the progressive shift from flexible external linguistic structure to compressed autonomous inner speech generation. This loss is independent of the diffusion policy $\pi$ and can be trained in a decoupled manner. Through this end-to-end optimization, the CVAE learns to fuse perceptual features, strategic intentions, and temporal context into compact representations—a process that connects to Vygotsky's concept of agglutination, where multiple semantic components bind into unified cognitive structures.

This training procedure implements a learning-based internalization process inspired by Vygotsky's theory of internalization. The VLM-generated descriptions serve as initial external linguistic targets, providing explicit structure that guides early training. Through continued training, the CVAE learns to autonomously generate compressed inner speech representations from behavioral patterns alone, implementing the structural transformation from elaborate external descriptions to the compact autonomous representations characteristic of mature inner speech.

**Algorithm 2 (Appendix A): Simulation.** During simulation, the agent must generate its own inner speech based on its ongoing behavior. To enable this, we use the CVAE decoder $\Psi_{\text{dec}}$ to generate the inner speech given the images of the past actions. We employ the W-step update cycle described in Section 3.1.3 for this purpose. We also wait for $t_0$ timesteps before the first generation. Thus, starting from a null speech of $m \leftarrow 0$, a new inner speech is generated periodically using $\Psi_{\text{dec}}$ after every $W$ timesteps starting from $t = t_0$. The framework enables explicit control over agent behavior through linguistic prompts, realizing the linguistic controllability aspect of the theoretical framework. By providing a natural language description $\mathcal{B}$ of the desired behavior, we override the initial inner speech from $m \leftarrow \mathbf{0}$ to $m \leftarrow \mathcal{B}$. The agent then continues its periodic updates to maintain consistency with the trained inner motivation space.

# 4 Experiments

## 4.1 Setup

**Datasets.** We use three robotic control environments from the D3IL benchmark [21]: *Aligning, Sorting, and Stacking*. These datasets include images taken from a top camera and an in-hand camera

during the demonstration. For the human-AI coordination task, we use the *Overcooked dataset* [5] in three different layouts: *Cramped room, Coordination ring, and Asymmetric Advantages*. These include two agents (one with a green and one with a blue hat). Both vision-based observations (denoted as "-Vision") and feature-based observations are considered for the evaluation. For simulation, we consider the standard rollouts and trajectories in the D3IL benchmark, while for Overcooked we simulate 100 games to evaluate our agent.

**Methods.**[3] We denote the proposed method as **MIMIC** while the state-of-art baseline behavior cloner is denoted **BC**. Each uses the same diffusion-based architecture of DDPM-T for behavior policy.

**Implementation.** We use the Adam optimizer [22] with the learning rate tuned for each dataset (following Jia et al. [21]) to train both our CVAE-based inner speech generator and the diffusion-based behavior cloner. The other hyperparameters are searched for the best performance, with first update step $t_0 \in \{0, 1, W/2, W - 1\}$ and update window $W \in \{1, 10, 20, 50, 100\}$ for low horizon tasks and $W \in \{100, 200, 300\}$ for higher ones. The optimal random dropout probability for the diffusion model training is found to be optimal $\in \{0, 0.1\}$ depending on task, while the parameter $\beta = 0.1$ was found to be optimal through tuning for training the CVAE. To obtain the inner speech from the VLM, we fix the batch size of inner speech $B = 8$. Further details on the training, computation, and other experimental setup can be found in Appendix E.

**Metrics.** For control tasks, we use the standard metrics of the D3IL benchmark during simulation, *i.e.*, success rate or proportion of successful completions of the task, and behavioral entropy comparing the entropy of simulations with a categorical distribution of behavior descriptors. Higher values indicate accurate learning of the successful and diverse behaviors shown in human demonstrations. To assess our objective of high fidelity imitation, we also report the mean distance from the end-point in the Aligning task, and the Wasserstein distance between the generated and training state and completion time distributions wherever feasible, following [33]. For Overcooked, we evaluate the performance using the mean collective reward obtained when the trained agent plays along with a proxy human agent and Wasserstein distance between the discrete actions.

## 4.2 Results

### 4.2.1 Does MIMIC achieve high fidelity imitation of diverse behaviors on D3IL benchmark?

Table 1: Comparison of MIMIC against BC with the DDPM-T architecture on the D3IL benchmark. '-wass' denotes Wasserstein Distance metrics for the non-vision environments (infeasible in case of vision) where state Wasserstein distance is calculated using 5 random rollouts.

| Environment | Model | Success rate ↑ | Distance ↓ | Entropy ↑ | State-wass ↓ | Time-wass ↓ |
|---|---|---|---|---|---|---|
| Aligning | BC | 0.6645 | 0.1105 | 0.4743 | 0.6961 | 59.034 |
| | MIMIC-S | **0.8021** | **0.0664** | 0.4184 | **0.0459** | 50.569 |
| | MIMIC-E | 0.7229 | 0.0847 | **0.6148** | 0.0492 | **45.397** |
| Aligning-Vision | BC | 0.1833 | 0.1875 | 0.0895 | - | - |
| | MIMIC-S | **0.2229** | 0.1885 | 0.0849 | - | - |
| | MIMIC-E | 0.2083 | **0.1849** | **0.1473** | - | - |
| Sorting-Vision | BC | 0.7972 | - | 0.3596 | - | - |
| | MIMIC-S | **0.8417** | - | 0.3719 | - | - |
| | MIMIC-E | 0.8083 | - | **0.4494** | - | - |
| | | 1 box / 2 box | - | 1 box / 2 box / 3 box | | |
| Stacking | BC | 0.8027 / 0.4879 | - | 0.2058 / 0.1503 / **0.1049** | 9.43 | **336.51** |
| | MIMIC-S | 0.8129 / **0.6074** | - | 0.1774 / 0.0737 / 0.0394 | **0.75** | 345.14 |
| | MIMIC-E | **0.8213** / 0.5333 | - | **0.2115** / **0.1556** / 0.0878 | 13.69 | **336.51** |

Table 1 shows that MIMIC is superior at generating human-like behaviors across four different benchmark D3IL environments, achieving higher entropy and higher success rate in its generated trajectories than the state-of-the-art DDPM-T-based behavioral cloner. We consider two different variants of MIMIC: *MIMIC-S and MIMIC-E*, denoting the combination of hyperparameters that gives the highest success rate and highest entropy, respectively. In almost all cases, both variants of MIMIC improve the performance over the BC model. We improve significantly in the Aligning task, while in

---

[3]We provide comparisons with additional behavior cloning approaches such as BESO [36] and BeT [39] in Appendix G.

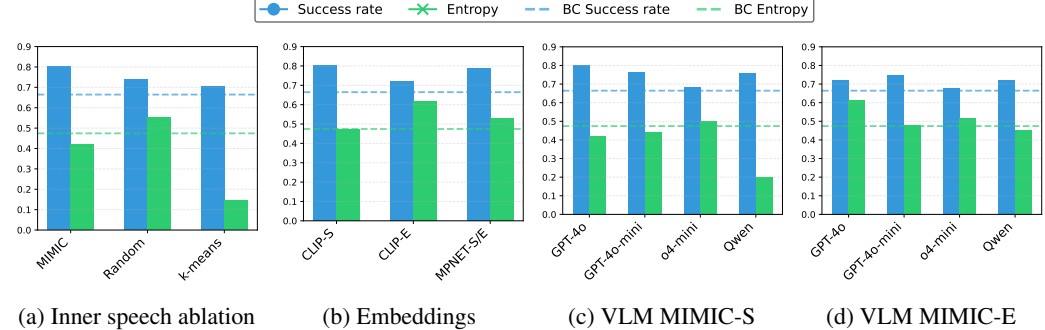

|   |   |   |   |
|---|---|---|---|
| (a) Inner speech ablation | (b) Embeddings | (c) VLM MIMIC-S | (d) VLM MIMIC-E |

Figure 4: Effect on Success Rate and Entropy performance of MIMIC on changing various components on the Aligning dataset: (a) removing inner speech during simulation, (b) changing the embedding model, (c-d) using different VLMs for training and scaffolding.

the more complex Stacking task, MIMIC-E achieves the best success rates and entropy for 1 and 2 boxes. increasing the success rate by 2 substantially. BC gives competitive values of entropy for the 3rd box, perhaps due to the actions being random since the success rate for the two boxes remains low. The gains of MIMIC remain consistent even in environments with vision-based observations, indicating that useful supplementary information is provided through inner speech. We further validate MIMIC's effectiveness in capturing diverse human behavior with high fidelity, showing significantly lower Wasserstein distance between the generated and training distributions of the state and completion time distributions (following [33]) in both aligning and stacking environments.

### 4.2.2 Do MIMIC agents serve as effective *in silico* human surrogates, achieving successful coordination when evaluated with human proxy models in collaborative tasks?

Table 2 shows that MIMIC significantly improves the collective reward achieved in three different settings of the Overcooked environment. The total reward achieved by the actions of MIMIC with the human agent is significantly higher than BC with the human agent in all cases, achieving an increase of up to 30. These results highlight the impact of using inner speech in improving human-AI coordination and demonstrate that modeling the inner speech of the agent consistently improves performance when evaluated against human proxy models. Additional results are provided in Appendix G.

Table 2: Comparison of MIMIC against BC with DDPM-T on the Overcooked environments.

| Environment | Model | Collective reward |
|---|---|---|
| Cramped room | BC | $115.8 \pm 3.86$ |
|  | MIMIC | $\mathbf{151.8 \pm 2.45}$ |
| Cramped room-Vision | BC | $73.6 \pm 6.18$ |
|  | MIMIC | $\mathbf{108.8 \pm 4.84}$ |
| Coordination ring | BC | $113.0 \pm 2.21$ |
|  | MIMIC | $\mathbf{121.0 \pm 1.93}$ |
| Asymmetric advantages | BC | $215.8 \pm 3.04$ |
|  | MIMIC | $\mathbf{227.6 \pm 2.69}$ |

### 4.2.3 How does the performance of MIMIC vary with different components?

**Ablation on inner speech.** To validate the importance of language as inner speech we compare MIMIC with other forms of inner speech such as a completely random vector and a clustered vector of the training trajectories. For clustering, we employ a *K-means algorithm* with $K = 8$ and learn a CVAE to generate the mean cluster representation for each action in the training set. These inner speech generators are used directly during simulation as described above. Figure 4a shows that MIMIC is significantly better than other formulations, with the worst being the K-means algorithm and random more effective than the base BC model.

**Embeddings.** Figure 4b shows the effect of changing the CLIP model to encode the linguistic descriptions with a text-only MPNET model [4]. We find that MIMIC-S with CLIP outperforms the MPNET variant in success rate, while MIMIC-E with CLIP outperforms it in entropy. This shows that a shared vision-language representation space is useful to obtain effective inner speech.

---

[4]https://huggingface.co/sentence-transformers/all-mpnet-base-v2

Table 3: GPT-4o evaluation. '-' denotes no update.

| $t_0$ | $W$ | Success rate | Entropy | GPT-4o Eval (1-5) | |
|---|---|---|---|---|---|
| | | | | Top | Inhand |
| *Training conditions* | | | | | |
| - | - | 0.6083 | 0.4690 | **4.03 ± 0.63** | **4.05 ± 0.51** |
| 10 | 20 | **0.7417** | 0.3815 | 3.98 ± 0.68 | 3.98 ± 0.81 |
| 49 | 50 | 0.6042 | **0.5156** | 3.88 ± 0.72 | **4.05 ± 0.71** |
| *Validation conditions* | | | | | |
| - | - | 0.6687 | 0.5787 | 3.83 ± 0.69 | **4.40 ± 0.49** |
| 10 | 20 | 0.6479 | 0.2949 | 3.94 ± 0.64 | 4.17 ± 0.37 |
| 49 | 50 | **0.7250** | **0.6357** | **4.23 ± 0.42** | 4.30 ± 0.46 |

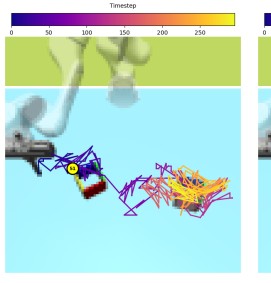 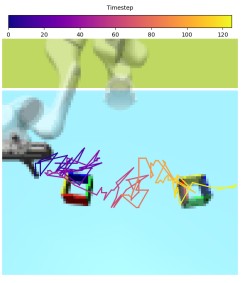

Grip the box from the right and rotate it counter-clockwise to achieve alignment

I approach from a diagonal top-right angle, requiring a swift rotation to align

Figure 5: Conditional behavior generation. Figures in the right panel show generated behaviors for the specified condition as inner speech with no subsequent updates.

**Vision-language model.** We study the effect of changing the VLM from GPT-4o to o4-mini, 4o-mini, and Qwen (Qwen2.5-VL-72B-Instruct). Figures 4c and 4d show that GPT-4o gives the best success rate compared with other VLMs, followed closely by the open-source Qwen model. Surprisingly, we find o4-mini to lack in success rate even though its descriptions lead to the most diversity (highest entropy). We believe this is due to the CLIP model failing to distinguish the nuanced behaviors generated using o4-mini. More importantly, we find that in all cases, MIMIC outperforms the BC model in success rate and often increases the base entropy in all cases except Qwen. This is likely due to a lack of diversity in Qwen generated descriptions of the dataset.

### 4.2.4 Can MIMIC be used to generate desired behavior, thereby enabling steerable imitation?

We study the effectiveness of MIMIC in generating desired behaviors. Here, we consider using the behavioral descriptions generated by the VLM for the training and validation sets as the desired conditions. Then, we follow Section 3 to give the desired condition as the initial inner speech and generate successful behaviors that can match the desired description. We then use GPT-4o as a judge to evaluate how well the generated trajectory matches the desired description (full prompt provided in Appendix E). Table 3 shows that MIMIC can be used to generate successful and controllable behaviors in the Aligning dataset. Using three different combinations of simulation parameters, we find that MIMIC generates highly successful and desired trajectories for both training and validation descriptions as conditions. Generating training behaviors is likely to need no mediation since behaviors are already conditionally captured, so no updates work best for training, while periodic updates help in validation. Figure 5 further shows examples of generated behavior with no periodic updates. In the first case, we note how it grips the box from the right side and keeps rotating it to follow the condition, while in the second case, we find an approach from a diagonal top-right angle for alignment. More examples are provided in Appendix G.

## 5 Conclusion

This paper introduces MIMIC, a framework that bridges cognitive science and imitation learning by operationalizing the theory of inner speech as a mediational mechanism between perception and action. By formalizing inner speech as a behavior guide, we address fundamental limitations in conventional behavior cloning approaches, which attempt to directly map states to actions. The empirical results validate the theoretical proposition that language-based internal representations enable more faithful modeling of human decision-making, shown here to achieve superior imitation fidelity with higher entropy and success rates while enabling designer-specified control. Appendix B provides an extended discussion on future directions, limitations, and the broader societal impact of our approach. Our findings establish the theoretical and practical viability of inner speech mechanisms as a computational foundation for imitation learning systems that can simultaneously exhibit behavioral richness and remain controllable. This research further opens significant avenues for investigation on potentially transforming how AI systems internalize human-like decision processes while establishing new research trajectories in language-mediated control and multi-agent collaboration—laying essential groundwork for systems that reliably collaborate across the full spectrum of human behaviors.

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

# Supplementary Material for Inner Speech as Behavior Guides

## A    Algorithms for MIMIC Training and Simulation

This section provides the pseudocode for the algorithms described in Section 3.2.2.

---

**Algorithm 1 MIMIC: Training**

**Require:** Set of human demonstrations $\mathcal{D}$, Batch size $B$, Hyperparameters for training
1: Obtain demonstration $\in \mathcal{D}$ as images $\mathbf{I}_{1:T}^{(i)}$.
2: Inner speech $[m^{(i)} \cdots m^{(i+B)}] \leftarrow$ CLIP $(\text{VLM}([\mathbf{I}_{1:T}^{(i)}, \cdots, \mathbf{I}_{1:T}^{(i+B)}]))$.
3: Construct $\mathcal{D}_M$ by augmenting $\{m^{(i)}\}$ to $\mathcal{D}$.
4: Train $\pi_\theta, \Psi$ to minimize $\mathcal{L}_{\text{diff}}(\mathcal{D}_M)$ [Eq. 3] and $\mathcal{L}_{\text{is}}(\mathcal{D}_M)$ [Eq. 4] respectively.

---

**Algorithm 2 MIMIC: Simulation**

**Require:** Initial state $s_1$ of the environment, First update step $t_0$, and update window $W$
1: Inner speech $m \leftarrow \mathbf{0}$
2: **for** $t = 1$ to T **do**
3:     **if** $t \pmod{W} \equiv t_0$ **then**
4:         Past images $\mathbf{I}_{t-W:t}$ from $s_{t-W:t}$
5:         Sample $z \sim \mathcal{N}(0, I)$.
6:         Update $m \leftarrow \Psi_{\text{dec}}(z, \mathbf{I}_{t-W:t})$
7:     **end if**
8:     Generate the action $a_t \sim \pi_\theta(\cdot \mid s_t, m)$
9:     Update the state $s_{t+1} \leftarrow \mathcal{E}(s_t, a_t)$.
10: **end for**

---

## B    Discussions

### B.1    Inner Speech as a Behavior Guide

MIMIC provides an alternative to the conventional behaviorist framework in IL in the form of a mediated action selection framework that is grounded in cognitive science. The fundamental distinction between the behaviorist and cognitive approaches to IL, as illustrated in Figure 6, represents more than a technical architectural choice. It reflects competing theories of how intelligent behavior emerges.

The behaviorist paradigm (left panel) conceptualizes human action as direct responses to environmental stimuli, where an agent learns a mapping function from states to actions $(s_t \mapsto_\mathcal{H} a_t)$. This approach, while computationally elegant, treats the human mind as a black box, assuming that behavioral patterns can be fully captured through observed input-output pairs. In contrast, the cognitive approach (right panel) recognizes that human actions are mediated by internal mental processes—what cognitive science literature terms 'inner speech,' internalized linguistic structures that guide behavior. Here, the same environmental state can produce diverse actions because it is filtered through an intermediate cognitive layer $(s_t \mapsto m_t \mapsto a_t)$, where m represents the inner dialogue that shapes interpretation and response selection. This mediational architecture explains a fundamental observation about human behavior: why different individuals, or even the same individual at different moments, can respond differently to identical situations.

In this way, the cognitive model seeks to capture not just what humans do but to also approximates how they deliberate, through internal linguistic reasoning that weighs options, considers context, and reflects individual motivations. The computational instantiation of this cognitive framework leverages the latent space of inner speech as a principled mechanism for both behavioral diversity and designer control. The continuous latent representation $m$ enables stochastic sampling during inference, naturally inducing behavioral variability that mirrors human decision-making heterogeneity, while simultaneously providing a semantic interface for control—designers can specify desired behaviors through natural language that constrains the latent distribution. Crucially, the vision-language model (VLM) shown in Figure 6 serves as developmental scaffolding, transforming visual observations into external linguistic descriptions that bootstrap the inner speech generator during training.

For artificial agents intended to collaborate with humans, the distinction between these two frameworks is critical: while behaviorist approaches may achieve high task performance, they fail to

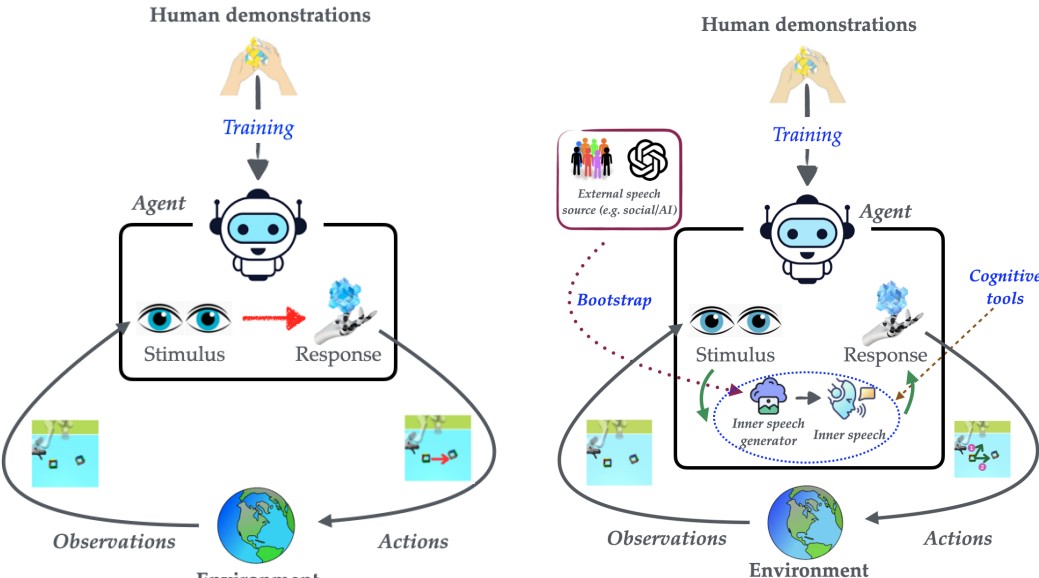

(a) Behaviorist framework: Direct stimulus-response mapping

(b) Cognitive framework: Linguistically-mediated action selection

Figure 6: Contrasting theoretical frameworks for IL. (a) The behaviorist approach models human behavior as a direct mapping from environmental states to actions ($s_t \mapsto_{\mathcal{H}} a_t$), treating cognitive processes as opaque transformations. (b) The cognitive approach instantiated by MIMIC introduces inner speech as a mediational layer ($s_t \to m_t \to a_t$), where $m_t$ represents linguistically-structured internal deliberation that enables behavioral diversity and contextual adaptation.

| Approach | Behavioral Diversity | Designer Control | Speech Type | Language Grounding | Language Annotations Required | Latent Space | Conditioning Type |
|---|---|---|---|---|---|---|---|
| Behavior Transformer [39] | ✓ (discrete modes) | × | N/A | × | **No** | × | Unconditional |
| Diffusion BC [33] | ✓ (continuous) | × | N/A | × | **No** | ✓ (implicit) | Unconditional |
| BESO [36] | ✓ (partial)[1] | ✓ (goals only) | External | ✓ (goals) | **Yes** (goals) | ✓ | Goal-conditioned |
| Thought Cloning [18] | ×[2] | × | External[3] | ✓ (thoughts) | **Yes** (per-step) | × | Unconditional |
| **MIMIC (Ours)** | ✓ (stochastic) | ✓ (general) | Internal | ✓ (inner speech) | **No**[4] | ✓ (CVAE) | General linguistic |

Table 4: Comparative analysis of IL approaches across key dimensions of behavioral modeling and control. **Legend:** ✓ = Full support; × = No support. **Annotations:** [1]Generates diversity through diffusion but only within goal-constrained trajectories. [2]Conditions on fixed human-provided thoughts, limiting emergent diversity. [3]Uses linguistic signals that remain external annotations rather than internally generated mediators. [4]Bootstraps from visual observations using VLM-generated captions, eliminating the need for human language annotations.

generate the behavioral diversity and contextual adaptability that characterize human partners, limiting their effectiveness in real-world collaborative scenarios where understanding and predicting varied human responses is essential. Table 4 distills MIMIC's unique position as the only approach that achieves language-grounded control without requiring exhaustive human linguistic supervision, instead leveraging vision-language models to automatically generate the necessary training signals from existing visual demonstrations.

## B.2 Limitations and Future Extensions

While MIMIC represents a significant advance in cognitively-grounded IL, several limitations warrant consideration for robust deployment across diverse contexts. First, the fidelity of inner speech generation remains contingent upon the quality of linguistic annotations produced by vision-language

models during training, creating a dependency where advances in behavioral modeling are partially gated by progress in vision-language understanding—though notably, improvements in VLM capabilities will naturally enhance MIMIC's performance without architectural modifications.

Second, the temporal granularity of inner speech generation, controlled through the $W$ parameter, requires careful calibration for different task domains, as excessive polling may induce behavioral instability through frequent re-planning while insufficient polling reduces the framework's capacity to correct for distributional drift.

Furthermore, while the CVAE's latent representation enables stochastic behavioral generation, the mapping between latent codes and semantic content remains opaque; post-hoc clustering or CLIP-based projection to natural language recovers partial interpretability but potentially loses nuanced behavioral intentions encoded in the continuous space.

Finally, the framework's efficacy in complex multi-agent environments with heterogeneous behavioral patterns remains unexplored, and coordination complexity may scale non-linearly with agent count in scenarios beyond dyadic interaction.

Future directions could include exploring semi-supervised approaches leveraging limited human annotations to calibrate VLM-generated captions so as to mitigate data quality dependencies, potentially through active learning frameworks that identify high-uncertainty trajectories for targeted human review. Adaptive polling mechanisms that dynamically adjust temporal granularity based on task complexity or behavioral uncertainty metrics would provide more robust default behaviors while reducing practitioner burden. Developing disentangled latent representations or incorporating discrete latent variables with explicit semantic grounding could enhance interpretability without sacrificing behavioral diversity. For multi-agent scalability, hierarchical inner speech architectures that model group-level intentions alongside individual cognition present a promising direction, enabling agents to reason about collective dynamics while maintaining individual behavioral authenticity.

### B.3   Beyond diffusion based behavior policy

While our current implementation employs a diffusion-based behavior policy (DDPM-T), the MIMIC framework is fundamentally model-agnostic. The core insight—that inner speech serves as a stochastic mediator between perception and action—can be instantiated with any conditional behavior cloning architecture. The framework decomposes into two independent components: (1) an inner speech generator $p(m|\mathcal{H}_t)$ that produces linguistic representations from behavioral history, and (2) a behavior policy $p(a|s, m)$ that conditions on these representations. This modular design means the behavior policy can be implemented using transformers (e.g., Behavior Transformer), flow-based models, energy-based models, or even standard supervised learning approaches—any architecture capable of conditional generation.

The key requirement is that the base model accepts an additional conditioning signal. Since inner speech is represented as a continuous embedding $m \in \mathcal{Z}$, it can be naturally incorporated through concatenation with state features, cross-attention mechanisms, FiLM conditioning, or additive conditioning depending on the architecture.

The periodic generation mechanism (parameter $W$) is similarly architecture-agnostic, as it operates at the simulation level rather than within the model architecture. Any autoregressive policy can maintain a fixed inner speech representation for $W$ steps before regenerating, making this a general inference-time control mechanism applicable across model families. Future work could explore this architectural flexibility to identify optimal policy architectures for different task domains while maintaining the core inner speech framework.

### B.4   Cognitive Inspiration vs. Biological Plausibility

Our framework draws computational inspiration from cognitive theory without claiming biological fidelity or neurobiological correspondence. This distinction is crucial: we operationalize functional properties from cognitive theories of inner speech—semantic condensation, predicativity, and temporal regulation—through computational mechanisms (CVAE, transformer attention, diffusion policy) rather than attempting to replicate neural substrates.

This approach follows a productive tradition in AI where psychological theories inform architectural design without requiring neural isomorphism. Convolutional networks leverage principles of hierarchical visual processing without mimicking V1 neurons; attention mechanisms capture aspects of human focus without replicating neural attention circuits. Similarly, MIMIC extracts functional principles from inner speech theory to address behavioral diversity in imitation learning.

Our technical contributions lie in: (1) formalizing inner speech properties through information-theoretic and probabilistic frameworks (Section 3.1), (2) instantiating these properties through specific architectural choices (Section 3.2), and (3) empirically validating that these computational mechanisms improve behavioral fidelity and diversity. We make no claims about whether artificial agents experience phenomenological "inner speech" or whether our architectures replicate human cognitive processes at a mechanistic level.

The value of cognitive inspiration lies in generating testable hypotheses about computational mechanisms—in our case, that introducing linguistic mediation between perception and action can capture human behavioral diversity. Our empirical results validate this computational hypothesis while remaining agnostic about biological implementation.

## B.5 Broader Impact

The development of cognitively-grounded artificial agents through MIMIC presents opportunities as well as ethical considerations for human-AI collaboration. The capacity to generate behaviorally-realistic human surrogates enables comprehensive pre-deployment safety validation, potentially preventing harmful interactions in high-stakes domains such as healthcare and autonomous systems. By incorporating linguistically-mediated control mechanisms, MIMIC also enhances transparency in AI decision-making while modeling cognitive diversity through stochastic inner speech generation—facilitating more inclusive systems that account for varied cultural reasoning patterns and individual differences in collaborative scenarios.

However, the same sophistication in replicating human-like behavioral patterns also introduces novel risks requiring careful governance. The ability to generate convincing behaviors could enable sophisticated social engineering attacks or deceptive AI personas designed to exploit human trust. When functioning correctly, such technology might enable unauthorized behavioral profiling; when producing incorrect outputs, it could generate inappropriate social behaviors violating cultural norms; and through intentional misuse, it could facilitate manipulative agents targeting human cognitive vulnerabilities. Additionally, biases embedded in vision-language models used for bootstrapping could perpetuate societal inequities if generated inner speech reflects discriminatory patterns in training corpora.

Mitigation strategies should focus on balancing innovation with principles of transparency and accountability. Disclosure of artificial agency through technical markers in inner speech generation could prevent deceptive practices. Establishing auditable logs of cognitive mediation processes would enable post-hoc analysis supporting accountability in high-stakes applications. These kinds of mitigations would would help to promote further advances in cognitively-grounded AI enhancing rather than undermining human agency in increasingly automated societies.

## C   Extended Related Work

**Imitation learning.**   IL algorithms are commonly organized into *behavior cloning*, *inverse-reinforcement learning*, and *distribution-matching* families [20, 32, 14]. Classic behavior cloning (BC) regresses actions from expert states; the seminal *ALVINN* system kept a vehicle in its lane by copying recorded steering commands [34]. Covariate-shift issues in BC motivated Dataset Aggregation (DAgger) [38], which iteratively queries experts on states visited by the learned policy to mitigate distribution mismatch. *Inverse-reinforcement learning (IRL)* infers a reward explaining the demonstrations [31, 1], while *generative adversarial IL (GAIL)* matches occupancy measures via an adversarial game [15]. The discriminator in GAIL can be interpreted as a potential-based reward, linking it back to IRL [12]. *Hierarchical IL* discovers latent sub-policies that can be sequenced for long-horizon manipulation [11]. While these approaches model imitation as direct mappings from states to actions or through inferred rewards, MIMIC introduces inner speech as a mediational

mechanism between perception and behavior, enabling both distributional matching and designer control through linguistic intervention—a capability absent in traditional IL paradigms.

**Diverse behavior imitation: from single mode to multimodal.** Human demonstrations are multi-modal. Recognizing this, InfoGAIL augments GAIL with an information term so a latent captures hidden styles [25], employing mutual information maximization to discover discrete behavioral modes within expert demonstrations.

Variational methods [45] learn conditional VAEs to embed motor skills, allowing one-shot imitation by sampling different latent states. Such approaches encode behavioral diversity through continuous latent representations that can be manipulated to generate novel skill variations. A hierarchical VAE extends this idea to multi-scale variation [11], decomposing complex behaviors into temporal hierarchies where higher-level latent representations control long-horizon strategies while lower-level latent representations capture execution details. Sequence models such as Behavior Transformers okenize continuous actions and clone $k$ distinct modes using prompt tokens [39], leveraging the transformer architecture's capacity for in-context learning to capture multimodal action distributions through discrete behavioral prototypes.

Score-based approaches fit diffusion models directly on trajectories, reproducing the full joint-action distribution [33]. These methods model the entire behavioral manifold through iterative denoising processes, achieving high-fidelity reproduction of demonstration diversity. BESO shows the same mechanism can be goal-conditioned with only three denoising steps [36], demonstrating computational efficiency while maintaining distributional expressiveness through accelerated diffusion sampling. These techniques broaden behavioral variety, but explicit control over which behavior type will appear at test time remains limited. This is a gap addressed by MIMIC's language-grounded inner speech. MIMIC builds on diffusion BC but conditions the denoising step on an inner-speech vector learned via a vision–language scaffold, enabling fine-grained linguistic control without retraining.

**Imitation learning for studying Human–AI coordination.** The deployment of AI agents in collaborative human environments necessitates computational approaches that transcend purely algorithmic optimization to encompass the full spectrum of human behavioral patterns and coordination dynamics. In multi-agent coordination domains, empirical evidence demonstrates the critical role of human behavioral modeling: in *Overcooked*, self-play agents confuse human partners due to convergence in self-play to non-human equilibria, whereas agents fine-tuned after first imitating human gameplay coordinate effectively [5]. Similarly, in *Hanabi*, monte-carlo search regularized with a human-behavior prior achieves high human-partner win rates by incorporating human-like suboptimalities and communication patterns. These systems employ a two-stage pipeline—learn a human model, then train a best response—yet this segregation introduces computational inefficiency and potential misalignment between the human model and the coordination policy.

The effectiveness of the above approaches is constrained by data availability: existing datasets exhibit significant limitations in capturing behavioral diversity. D4RL offers benchmark tasks but its demonstrations stem from synthetic experts, limiting stylistic variety [13]; RoboNet amasses large tele-operation sets yet focuses on narrow table-top primitives [8]; CALVIN provides language supervision but shows a single canonical solution per goal [26]; and Overcooked traces are short and stylistically homogeneous [5]. D3IL deliberately captures multiple human strategies for each manipulation task, making it a rare test-bed for diversity [21]. The BabyAI dataset is another exception, in providing explicit thought annotations for every action, enabling *thought cloning* [18] to learn from paired action-thought demonstrations. However, this kind of linguistic supervision is expensive and its availability is rare.

MIMIC addresses the architectural and data challenges: it unifies the two-stage pipeline as the diffusion policy trained by imitation already exhibits human-like variability and can serve directly as the partner model during new-agent optimization, while mitigating data bottlenecks by using automatically generated captions to train its inner-speech generator on existing video-only corpora—effectively bootstrapping linguistic mediation from visual demonstrations without requiring the exhaustive human annotation.

**Language Interfaced Imitation Learning.** *Thought cloning (TC)* [18] discussed above directly imitates human thoughts but requires access to annotation of actual human thought for each step in the demonstration trajectory. Further, the performance of TC is highly tied with goal (mission) conditioning and degrades by almost 40% in our experiments once the goal (mission) condition

is removed. Similarly, external speech has been used to steer agent behaviors through action re-ranking [30], though these speech mechanisms remain external to the agent. [43] enforce linguistic bottlenecks through auxiliary tasks but through approaches that operate outside the IL paradigm. Closest prior is [46], who frame intra-agent speech as *semi-supervised captioning*: a vision-language captioner is pretrained and then *frozen* to provide auxiliary caption and caption-matching supervision that improves behavior cloning and enables zero-shot object-level generalization with few additional captions. In contrast, we model *inner speech* as an explicit *latent mediator* that *conditions* the policy, i.e., $p(a \mid s) = \int p(a \mid s, m)\, p(m \mid s)\, dm$, and we *generate $m$ online* from recent history via a conditional VAE. This shift from auxiliary language supervision to mediational control yields *steerable* and *distributionally realistic* imitation (designer-prompted behaviors, periodic refresh) rather than only improved supervision signals.

**Cognitive Theories of Inner Speech and Behavioral Diversity.** The relationship between inner speech and behavioral diversity has been extensively studied within cognitive psychology and neuroscience, providing rich theoretical foundations for our computational approach. Vygotsky's seminal work [44] established inner speech as internalized social dialogue that mediates higher cognitive functions, proposing that the transformation of interpersonal communication into intrapersonal dialogue creates a mechanism for behavioral self-regulation. This theoretical framework was subsequently empirically observed by Sokolov [42], who characterized inner speech as possessing distinctive structural and functional properties that differentiate it from external communication.

Contemporary cognitive research has extended these foundations to explain behavioral diversity. Fernyhough's [10] *dialogic theory* posits that inner speech maintains the dialogical characteristics of interpersonal communication, suggesting that behavioral diversity emerges from the multiplicity of internalized perspectives. As Alderson-Day and Fernyhough [2] note, "inner speech allows for the simulation of multiple action pathways before behavioral execution," providing a cognitive mechanism for generating diverse behavioral responses to identical environmental stimuli.

Neuroimaging studies [3] have identified neural correlates of inner speech, showing activation in both language production regions and motor planning areas during inner speech episodes. These findings are consistent with the hypothesis that inner speech serves as a cognitive rehearsal mechanism for behavioral alternatives. Morin's [29] self-regulatory framework further proposes that inner speech functions as a behavioral selection mechanism, where verbalized thoughts act as "cognitive filters" that modulate action selection based on contextual factors beyond immediate environmental stimuli.

This theoretical perspective aligns with empirical observations in human imitation learning. Meltzoff's [27] "like me" framework demonstrates that human imitation is not merely mimicry but rather an inferential process that reconstructs the intentions and mental states underlying observed actions. The diversity in imitative behavior derives from this reconstructive process, where different individuals generate different internal models of the demonstrator's cognitive states. Our computational architecture operationalizes this cognitive process, modeling how inner speech mediates between observation and action to produce diverse yet contextually appropriate behaviors.

**AI Approaches to Modeling Cognitive Processes.** Some AI research has explored computational implementations of cognitive processes. [6] propose "autotelic AI" that internalizes language for self-directed learning, emphasizing language as a tool for goal generation and intrinsic motivation. While sharing our interest in cognitive foundations, autotelic systems focus on autonomous learning rather than behavioral diversity, using language primarily for goal generation. [19] utilize large language models to simulate reasoning processes ("chain of thought") before action selection, implementing serial, deterministic reasoning rather than the stochastic, parallel processing characteristic of inner speech in Vygotskian theory. Our framework models inner speech as a probabilistic process generating diverse behavioral patterns from identical environmental states, more closely aligning with cognitive theories of human behavioral diversity. [4] propose natural language as a latent space for reinforcement learning, using language to structure behavior hierarchically. While this approach shares with us the adoption of language as a cognitive tool, it focuses on decomposing complex tasks rather than generating behavioral diversity. Our framework uniquely combines the stochastic nature of inner speech with IL to capture a wide spectrum of human behavioral variation without requiring explicit linguistic supervision.

# D  Technical Background

In this section, we provide a detailed background on diffusion models and conditional variational autoencoders, which constitute the backbone of MIMIC's architecture.

## D.1  Diffusion Models

Diffusion models, specifically *denoising diffusion probabilistic models (DDPMs)*, constitute a class of generative models that transform noise distributions into target data distributions through iterative denoising processes. Their theoretical foundation derives from non-equilibrium thermodynamics and Markovian diffusion processes, establishing a principled approach to generative modeling through progressive noise injection and removal [17, 9, 40].

The diffusion framework comprises two fundamental stochastic processes operating in complementary directions. The **forward diffusion process** defines a Markov chain that incrementally incorporates Gaussian noise according to a predefined variance schedule, systematically destroying the data structure. Given data $\mathbf{x}_0 \sim q(\mathbf{x})$, this process generates increasingly noisy latents through:

$$q(\mathbf{x}_t|\mathbf{x}_{t-1}) = \mathcal{N}(\mathbf{x}_t; \sqrt{1 - \beta_t}\mathbf{x}_{t-1}, \beta_t\mathbf{I}), \tag{5}$$

where $\{\beta_t\}_{t=1}^T$ represents the noise schedule with $0 < \beta_t < 1$. The noise schedule can follow various strategies including linear, cosine, or learned schedules, each affecting the quality-efficiency trade-off during generation. This formulation admits a tractable closed-form expression for any timestep:

$$q(\mathbf{x}_t|\mathbf{x}_0) = \mathcal{N}(\mathbf{x}_t; \sqrt{\bar{\alpha}_t}\mathbf{x}_0, (1 - \bar{\alpha}_t)\mathbf{I}), \tag{6}$$

where $\alpha_t = 1 - \beta_t$ and $\bar{\alpha}_t = \prod_{i=1}^t \alpha_i$. As $T \to \infty$ with an appropriate schedule, $\mathbf{x}_T$ approximates an isotropic Gaussian distribution, effectively erasing all information about the original data.

The **reverse diffusion process** recovers the original data distribution through learned denoising transformations, parameterized as:

$$p_\theta(\mathbf{x}_{t-1}|\mathbf{x}_t) = \mathcal{N}(\mathbf{x}_{t-1}; \boldsymbol{\mu}_\theta(\mathbf{x}_t, t), \boldsymbol{\Sigma}_\theta(\mathbf{x}_t, t)), \tag{7}$$

where $p_\theta(\mathbf{x}_T) = \mathcal{N}(\mathbf{x}_T; \mathbf{0}, \mathbf{I})$. Following established practice, the variance is typically fixed as $\boldsymbol{\Sigma}_\theta(\mathbf{x}_t, t) = \sigma_t^2\mathbf{I}$, with $\sigma_t^2$ either learned or set to $\beta_t$ or $\tilde{\beta}_t = \frac{1-\bar{\alpha}_{t-1}}{1-\bar{\alpha}_t}\beta_t$. The mean $\boldsymbol{\mu}_\theta(\mathbf{x}_t, t)$ is parameterized through a neural network that predicts the noise component:

$$\boldsymbol{\mu}_\theta(\mathbf{x}_t, t) = \frac{1}{\sqrt{\alpha_t}} \left( \mathbf{x}_t - \frac{\beta_t}{\sqrt{1 - \bar{\alpha}_t}}\boldsymbol{\epsilon}_\theta(\mathbf{x}_t, t) \right), \tag{8}$$

where $\boldsymbol{\epsilon}_\theta(\mathbf{x}_t, t)$ predicts the noise added during the forward process. This parameterization establishes a fundamental connection to score-based generative models, as the predicted noise is proportional to the score function $\nabla_{\mathbf{x}_t} \log q(\mathbf{x}_t)$.

The training objective, derived from variational inference principles, minimizes the negative evidence lower bound (NELBO). However, empirical investigations demonstrate that a simplified objective yields superior practical results:

$$\mathcal{L}_{\text{simple}} = \mathbb{E}_{t\sim\mathcal{U}[1,T],\mathbf{x}_0\sim q(\mathbf{x}),\boldsymbol{\epsilon}\sim\mathcal{N}(\mathbf{0},\mathbf{I})} \left[ \|\boldsymbol{\epsilon} - \boldsymbol{\epsilon}_\theta(\sqrt{\bar{\alpha}_t}\mathbf{x}_0 + \sqrt{1 - \bar{\alpha}_t}\boldsymbol{\epsilon}, t)\|^2 \right]. \tag{9}$$

This formulation enables efficient training through direct noise prediction across all timesteps simultaneously, while sampling necessitates iterative denoising from pure noise.

## D.2  Conditional Variational Autoencoders

*Conditional variational autoencoders (CVAEs)* extend the traditional VAE framework by incorporating conditional information into the generative process, thereby enabling controlled generation based on specified attributes, contextual constraints, or structural specifications [41, 23]. This conditional paradigm addresses the fundamental limitation of standard VAEs in providing explicit control over generated outputs, establishing CVAEs as particularly valuable for applications demanding targeted generation capabilities.

CVAEs introduce a conditioning variable $\mathbf{c}$ to model the conditional data distribution $p(\mathbf{x}|\mathbf{c})$ through a latent variable framework. The generative process is formulated hierarchically as:

$$p_\theta(\mathbf{x}|\mathbf{c}) = \int p_\theta(\mathbf{x}|\mathbf{z}, \mathbf{c}) p_\theta(\mathbf{z}|\mathbf{c}) d\mathbf{z}. \tag{10}$$

The conditioning mechanism operates at multiple architectural levels: influencing the prior distribution $p_\theta(\mathbf{z}|\mathbf{c})$, the likelihood $p_\theta(\mathbf{x}|\mathbf{z}, \mathbf{c})$, or both components simultaneously. Different conditioning strategies yield distinct modeling capabilities, ranging from simple attribute control to complex structural generation tasks requiring sophisticated conditional dependencies.

Since direct computation of the posterior $p_\theta(\mathbf{z}|\mathbf{x}, \mathbf{c})$ remains intractable, CVAEs employ variational inference with an approximate posterior $q_\phi(\mathbf{z}|\mathbf{x}, \mathbf{c})$, yielding the conditional evidence lower bound (ELBO):

$$\log p_\theta(\mathbf{x}|\mathbf{c}) \geq \mathbb{E}_{q_\phi(\mathbf{z}|\mathbf{x}, \mathbf{c})} \left[\log p_\theta(\mathbf{x}|\mathbf{z}, \mathbf{c})\right] - D_{KL}\left(q_\phi(\mathbf{z}|\mathbf{x}, \mathbf{c})\|p_\theta(\mathbf{z}|\mathbf{c})\right). \tag{11}$$

The conditional prior $p_\theta(\mathbf{z}|\mathbf{c})$ can be parameterized as a learned function of the conditioning variable, enabling the model to adapt the latent space structure based on conditional information. This adaptability fundamentally distinguishes CVAEs from simpler conditional generation approaches that merely concatenate conditions with inputs.

Neural networks parameterize both the encoder $q_\phi(\mathbf{z}|\mathbf{x}, \mathbf{c})$ as a conditional Gaussian distribution and the decoder $p_\theta(\mathbf{x}|\mathbf{z}, \mathbf{c})$, which reconstructs inputs based on both latent variables and conditioning information. The encoder produces conditional distributional parameters:

$$q_\phi(\mathbf{z}|\mathbf{x}, \mathbf{c}) = \mathcal{N}(\mathbf{z}; \boldsymbol{\mu}_\phi(\mathbf{x}, \mathbf{c}), \text{diag}(\boldsymbol{\sigma}_\phi^2(\mathbf{x}, \mathbf{c}))). \tag{12}$$

The reparameterization trick facilitates gradient-based optimization through:

$$\mathbf{z} = \boldsymbol{\mu}_\phi(\mathbf{x}, \mathbf{c}) + \boldsymbol{\sigma}_\phi(\mathbf{x}, \mathbf{c}) \odot \boldsymbol{\epsilon}, \quad \boldsymbol{\epsilon} \sim \mathcal{N}(\mathbf{0}, \mathbf{I}). \tag{13}$$

The complete training objective minimizes the negative conditional ELBO:

$$\mathcal{L}_{\text{CVAE}} = -\mathbb{E}_{q_\phi(\mathbf{z}|\mathbf{x}, \mathbf{c})} \left[\log p_\theta(\mathbf{x}|\mathbf{z}, \mathbf{c})\right] + D_{KL}\left(q_\phi(\mathbf{z}|\mathbf{x}, \mathbf{c})\|p_\theta(\mathbf{z}|\mathbf{c})\right). \tag{14}$$

This objective balances reconstruction fidelity against latent space regularization while incorporating conditional constraints. The KL divergence term encourages the approximate posterior to remain proximate to the conditional prior, enabling meaningful interpolation within the conditional manifold and ensuring that latent representations respect the conditioning structure.

# E   Experimental Setup

## E.1   Additional Environment details

Figure 7 (a,b,c) illustrates the D3IL [5] environments used in our experiments. We use the 4-box and vision-based setting for the Sorting environment, as we observed more stability and higher performance in the BC model under these settings. The following outlines some brief details on those environments.

**Aligning:** The Aligning task requires the robot to precisely manipulate a box such that it aligns with a target box within specified tolerances, with the constraint that colors must match for each side. The task admits two distinct behavioral modalities: pushing from the inside or from the outside of the box configuration, thereby introducing controlled multi-modality in the action space. The state representation encompasses end-effector position in Cartesian space, pushing box position and quaternion, and target box position and quaternion, with actions represented as desired Cartesian velocities. This task exemplifies the challenge of precision control under multi-modal behavioral strategies, requiring policies to master fine-grained manipulation while maintaining behavioral diversity.

**Sorting:** The Sorting task requires the robot to sort red and blue blocks into their color-matching target boxes, with task complexity scaling from 2 to 6 blocks. For the 6-block variant, the task exhibits

---
[5] https://github.com/ALRhub/d3il (MIT License)

20 distinct behaviors and demands complex manipulation sequences with high variation in trajectory lengths, challenging existing IL approaches. The state representation includes end-effector position, all boxes' positions and tangent of Euler angles along the z-axis, with dimensionality scaling linearly with the number of objects. This environment tests an agent's capacity to handle combinatorial complexity and maintain closed-loop sensory feedback across extended manipulation sequences.

**Stacking:** The Stacking task requires the robot to sequentially stack 1-3 blocks in a designated yellow target zone, employing a parallel gripper and augmented reality control interface for enhanced dexterity. The state representation includes robot joint positions, gripper width, and boxes' positions with Euler angle tangents, while actions encompass both joint velocities and gripper width control. Success criteria demand not only lateral positioning within the target zone but also appropriate vertical heights confirming successful stacking. This task represents the pinnacle of manipulation complexity in the D3IL suite, requiring precise grasp-place sequences, dynamic stability maintenance, and adaptive recovery from perturbations.

Figure 7 (d,e,f) illustrates the Overcooked [6] environments used in our experiments. **Note:** We use the term "Greedy agent" to report results for Overcooked environments, however, this agent is the same as the human proxy agent (split of the trajectories collected from humans) as reported in [5].

**Cramped Room:** Agents must navigate a confined workspace while executing sequential cooking tasks. The constrained spatial topology induces frequent collision possibilities, necessitating real-time trajectory adaptation and implicit coordination protocols that emerge through embodied interaction rather than explicit communication. The environment's state space encompasses agent positions, object locations, and cooking progress indicators, with actions comprising discrete movement commands and object interactions. This layout operationalizes fundamental questions about emergent coordination strategies in spatially constrained multi-agent systems, where optimal policies must balance task efficiency against collision avoidance through anticipatory modeling of partner trajectories.

**Asymmetric Advantages:** The Asymmetric Advantages layout tests whether agents can develop high-level strategic reasoning that leverages differential access to resources, as players begin in distinct spatial regions with asymmetric proximity to cooking stations. This environmental structure necessitates role specialization and adaptive task allocation, where agents must infer and exploit comparative advantages based on spatial positioning and partner capabilities. The layout embodies game-theoretic coordination challenges where multiple Nash equilibria exist, each corresponding to different role assignments and workflow patterns. Success requires agents to transcend myopic task completion toward globally efficient coordination strategies that emerge through iterated interaction and mutual adaptation.

**Coordination Ring:** The Coordination Ring layout presents a topologically constrained environment where the ring-like spatial structure forces agents to establish and maintain directional conventions (clockwise or counterclockwise movement) to prevent deadlock scenarios. Agents must rapidly converge on shared behavioral protocols without explicit communication channels. The circular topology creates a coordination game with multiple equilibria, where misaligned conventions result in systematic inefficiencies through blocking behaviors. This layout thus serves as a minimal testbed for studying how artificial agents can develop and adapt to emergent social conventions, mirroring fundamental processes in human social coordination where arbitrary but stable behavioral patterns facilitate collective action.

### E.1.1 Environment descriptions for the inner speech prompt

**Aligning.** The GIF(s) show a {camera} view of your actions and your task was to move a box starting from different positions using a robotic hand to align with the other box, which is fixed.

**Sorting.** The GIF(s) show a {camera} view of your actions where your goal was to sort red and blue blocks to their color-matching target box.

**Stacking.** The GIF(s) show a {camera} view of your actions where your goal to stack blocks with different colors in a (yellow) target zone.

---

[6]https://github.com/HumanCompatibleAI/overcooked_ai (MIT License)

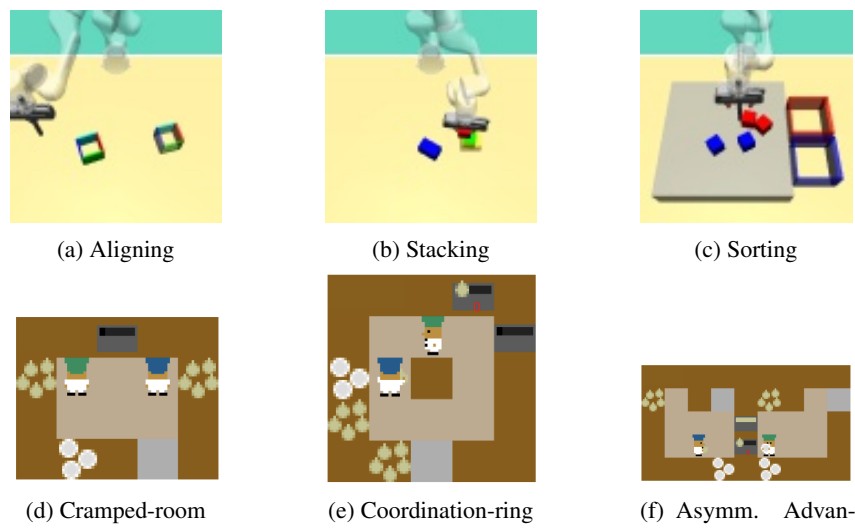

Figure 7: Environments used in our experiments. **(a-c)**: D3IL environments Aligning, Stacking and Sorting environments, respectively. **(d-f)**: Overcooked map layouts Cramped-room, Coorindation ring, and Asymmetric Advantage, respectively.

**Overcooked**: The goal is to place three onions in a pot (dark grey), take out the resulting soup on a plate (white) and deliver it (light grey), as many times as possible within the time limit. The GIF(s) show how the {agent} moves and interacts with other agents in a cramped room.

## E.2 Conditional evaluation

```
Evaluate the ability of the caption to describe the overall motion
in the gif from 1 to 5.
Also give your reasoning. The caption is deliberately succinct
and ignores the small motion so do not consider these points while
evaluating.
Caption: {caption}
GIF: attached
Score:
```

## E.3 Large language models

We use API version '2025-01-01-preview' for all models and employ structured output API to obtain the behavior descriptions corresponding to each GIF. [7]

## E.4 Computational environment

All experiments were conducted on a high-performance computing server, equipped with a 64-core x86_64 processor (128 threads) and 1007 GB of RAM, running Ubuntu 22.04 LTS (kernel 5.15.0-153-generic). For GPU-accelerated computations, we utilized an NVIDIA A100 80GB PCIe GPU with CUDA version 12.6. The experiments were implemented using Python 3.10.14 (conda-forge distribution) with PyTorch 2.7.0.

## E.5 Extension to multi-agent speech

A novel implication of MIMIC's flexibility arises in multi-agent settings where an agent can be conditioned not only on its own inner speech but also on the inner speech of other agents, inspired by

---

[7]https://platform.openai.com/docs/guides/structured-outputs

mirror neuron theory of social cognition [37]. This extension models how inner speech mediates social interaction, allowing agents to adapt their behavior based on their perception of others' cognitive states[8]. We denote it as MIMIC-MA in our experiments.

## F    Complexity Analysis

**Training.** First, generating inner-speech captions with GPT-4o is inexpensive: we make $N/B$ API calls with average context length of $\sim 100$ for text and $\sim B \cdot (85 + 170n)$ for images (with $n$ tiles of $512 \times 512$ px), totaling $\sim N \cdot 170n$ tokens which is about \$ 2 for over 400 trajectories, even with high resolution $2048 \times 2048$ images. The CLIP model ($\sim 0.5$ B parameters) is likewise lightweight and can be efficiently used during training to generate the inner speech.

**Inference.** Let $T_{CVAE}$ and $T_{diff}$ denote one forward pass through the CVAE and diffusion models, respectively. Over a simulation horizon $H$ with window size $W$, we perform $H$ diffusion passes and $H/W$ CVAE passes, yielding a total complexity of $O(HT_{diff} + H/WT_{CVAE})$. Since both are vision-conditioned with similar runtimes and $H > H/W$, the diffusion term dominates. So, MIMIC adds no inference overhead.

## G    Additional Experiment Results

We first report the extended results in each environment along with the parameter configurations that correspond to the reported best performance. We then analyze the sensitivity of MIMIC to various hyper-parameters and different VLM models. We conclude with the visualization of behaviors obtained through designer specified control text for the Aligning and Sorting environments.[9]

### G.1    How is inner speech represented in the embedding space?

Figure 8 shows the TSNE visualization of the CLIP-encoded inner speech of different environments, as generated using GPT-4o. We find that it tends to cluster together similar behaviors while separating distinct behaviors in this 2D space.

### G.2    Which configurations maximize the efficacy of MIMIC?

**Robotic Manipulation Task.** Table 5 reports the hyperparameters corresponding to the best performance reported in Table 1 for D3IL benchmark. Here, $p_{drop}$ denotes the probability of randomly dropping the $m$ for $\mathcal{L}_{\text{diff}}$. We note that higher update windows are preferred for long horizon environments, such as Stacking and Sorting than Aligning, where smaller update windows ($W$) gives high performance. Initial steps show more variation while indicating that higher values are preferred in non-vision environments. This means that for the first few steps, the agent takes its action with no inner speech. On the other hand, random dropout probability of inner speech ($p_{drop}$) is found to be important for the Aligning environment for higher performance while no such dropout is more useful in others.

**Overcooked Tasks.** Table 6 shows the hyperparameters along with the Wasserstein distance between the actions for the OverCooked environment. We also include the reward collected by the PPO$_{H_{proxy}}$ model from [5]. This approach trains the PPO agent in partnership with the $H_{proxy}$ model, essentially giving it access to ground truth. [5] calls this value the "gold standard" and reports only the PPO agent trained in the presence of an imitator reaching close to the gold standard. The results demonstrate that MIMIC already outperforms BC significantly and reaches closer to or surpasses the gold standard performance, demonstrating the capability of MIMIC agent to collaborate effectively with human proxy model. The results further show that the best hyperparameters often include a low initial step and a high update window with a non-zero dropping of probability. The Wasserstein distance between the generated and training actions to also small, following the trend in Table 1. This showcases the high fidelity of behavior imitation as compared to just task success that MIMIC achieves.

---

[8]In multi-agent settings, agents can observe each other's behaviors and infer corresponding inner speech representations, or share inner speech explicitly in cooperative scenarios where communication is available.

[9]For other environments, it was difficult to visualize the behaviors via static images and so we defer them to the gif versions shared along with the source code at our project website.

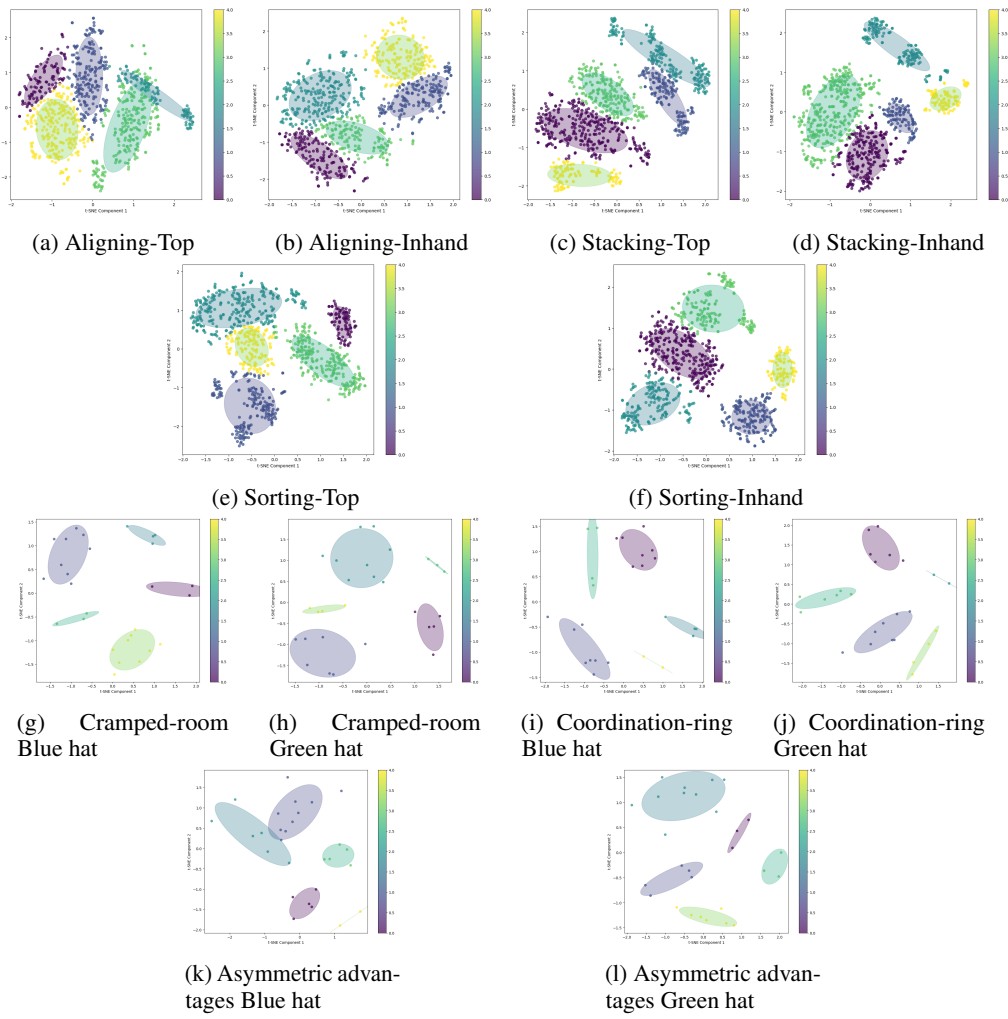

Figure 8: TSNE visualization of CLIP-encoded inner speech generated using GPT-4o for the environments used in our experiments. **(a-f)**: Top and inhand cameras in D3IL: Aligning, Stacking and Sorting environments, respectively. **(g-l)**: Blue and green hat agents in overcooked map layouts: Cramped-room, Coorindation ring, and Asymmetric Advantages, respectively.

We also study the setting of using the other agent's speech (denoted as MIMIC-MA) and find the performance to slightly improve in Coordination-ring, a layout where coordination becomes central to the task. Using the green hat agent's speech in this case is found to be useful, whereas in all other layouts, the agents collect less reward when using just the blue hat agent's own speech.

### G.3 How sensitive is MIMIC to hyperparameters?

Figure 9 shows how performance varies with change in the initial step and polling window of the simulation in the D3IL benchmark. The performance goes down by delaying the first step of the inner speech update. The performance improves when increasing the update window in the Aligning dataset, but only up to a limit after which the success rate starts going down while the entropy increases. We find that higher polling windows are preferred in Stacking and Sorting environments, while other trends are similar to Aligning.

Figures 10b and 10c shows the hyperparameter sensitivity in Overcooked cramped room environment and we find a similar trend as D3IL benchmark of increasing the performance with increase in the initial step to some extent, while update/polling windows show a drop in performance after a point.

Table 5: Comparison of MIMIC against BC with the DDPM-T architecture on the D3IL benchmark.

| Environment | Model | $p_{drop}$ | $t_0$ | $W$ | Success rate ↑ | Distance ↓ | Entropy ↑ |
|---|---|---|---|---|---|---|---|
| Aligning | BC | | | | 0.6645 | 0.1105 | 0.4743 |
| | MIMIC-S | 0.1 | 50 | 50 | **0.8021** | **0.0664** | 0.4184 |
| | MIMIC-E | 0.1 | 12 | 50 | 0.7229 | 0.0847 | **0.6148** |
| Aligning-Vision | BC | | | | 0.1833 | 0.1875 | 0.0895 |
| | MIMIC-S | 0.1 | 1 | 20 | **0.2229** | 0.1885 | 0.0849 |
| | MIMIC-E | 0.0 | 1 | 20 | 0.2083 | **0.1849** | **0.1473** |
| Sorting-Vision | BC | | | | 0.7972 | - | 0.3596 |
| | MIMIC-S | 0.0 | 100 | 200 | **0.8417** | - | 0.3719 |
| | MIMIC-E | 0.0 | 50 | 100 | 0.8083 | - | **0.4494** |
| | | | | | 1 box / 2 box | - | 1 box / 2 box / 3 box |
| Stacking | BC | | | | 0.8027 / 0.4879 | - | 0.2058 / 0.1503 / **0.1049** |
| | MIMIC-S | 0.0 | 30 | 50 | 0.8129 / **0.6074** | - | 0.1774 / 0.0737 / 0.0394 |

Table 6: Comparison of MIMIC against BC with DDPM-T on the Overcooked environments. '-' denotes "action Wasserstein" is not feasible or not available. * denotes values taken directly from [5]. Note that "state Wasserstein" is infeasible due to a large dimension (96) of state features.

| Environment | Model | $p_{drop}$ | $t_0$ | $W$ | Collective reward | Action Wasserstein |
|---|---|---|---|---|---|---|
| Cramped room | $PPO^*_{H_{proxy}}$ | | | | $\sim 155 - 160$ | - |
| | BC | | | | $115.8 \pm 3.86$ | 0.24 |
| | MIMIC | 0.1 | 10 | 100 | $\mathbf{151.8 \pm 2.45}$ | 0.25 |
| | MIMIC-MA | 0.1 | 1 | 50 | $148.4 \pm 2.17$ | 0.25 |
| Cramped room-Vision | BC | | | | $73.6 \pm 6.18$ | - |
| | MIMIC | 0.0 | 1 | 50 | $\mathbf{108.8 \pm 4.84}$ | - |
| | MIMIC-MA | 0.0 | 1 | 20 | $103.6 \pm 3.69$ | - |
| Coordination ring | $PPO^*_{H_{proxy}}$ | | | | $\sim 145 - 150$ | - |
| | BC | | | | $113.0 \pm 2.21$ | 0.08 |
| | MIMIC | 0.1 | 10 | 50 | $121 \pm 1.93$ | 0.09 |
| | MIMIC-MA | 0.1 | 10 | 20 | $\mathbf{128.6 \pm 1.75}$ | 0.03 |
| Asymmetric advantages | $PPO^*_{H_{proxy}}$ | | | | $\sim 125 - 130$ | - |
| | BC | | | | $215.8 \pm 3.04$ | 0.14 |
| | MIMIC | 0.1 | 10 | 200 | $\mathbf{227.6 \pm 2.69}$ | 0.10 |
| | MIMIC-MA | 0.1 | 10 | 50 | $227.0 \pm 1.84$ | 0.11 |

We also evaluate the effect of changing the embedding and VLM in the overcooked environment. Figure 10a shows that CLIP-encoded and GPT-4o-scaffolded inner speech is most effective in obtaining the highest collective reward in the Overcooked cramped room. However, we find that even by changing the embedding and VLM, MIMIC still outperforms the BC variant.

### G.4 How does MIMIC compare against other strong imitation learning approaches?

While our choice of BC (DDPM-T, [33]) is motivated by its high benchmark performance, we also compare against two additional approaches for comprehensiveness: BESO [36] and BeT [39]. For a fair comparison, we use the BESO's diffusion model architecture as the underlying policy network in MIMIC instead of a DDPM-T architecture. Table 8 shows that MIMIC substantially outperforms these approaches as well, further highlighting the advantages of using inner speech.

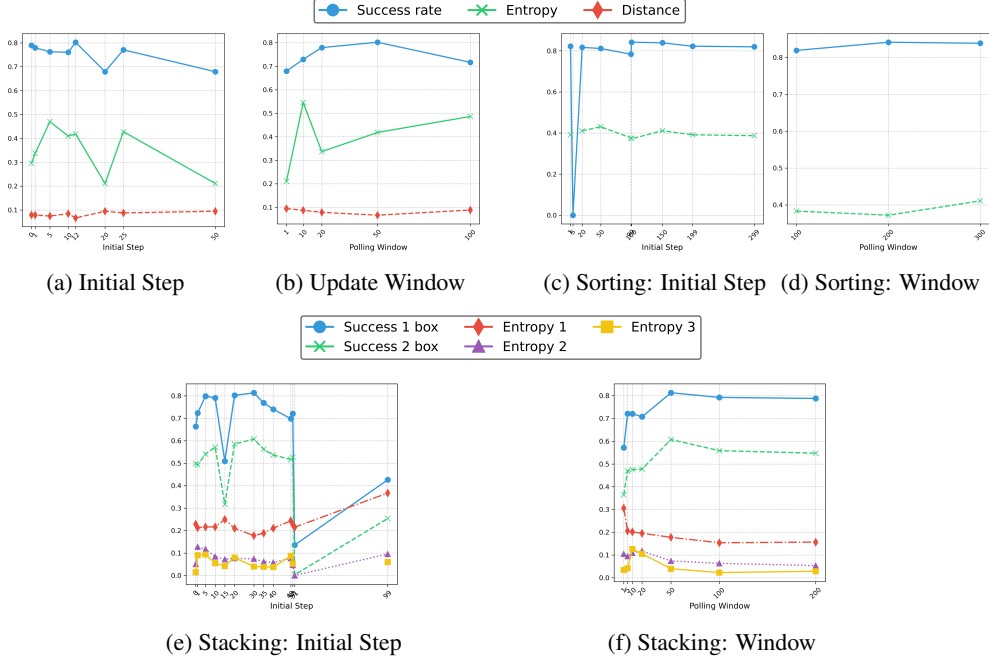

(a) Initial Step     (b) Update Window     (c) Sorting: Initial Step     (d) Sorting: Window

(e) Stacking: Initial Step           (f) Stacking: Window

Figure 9: Hyperparameter sensitivity of MIMIC on the D3IL benchmark.

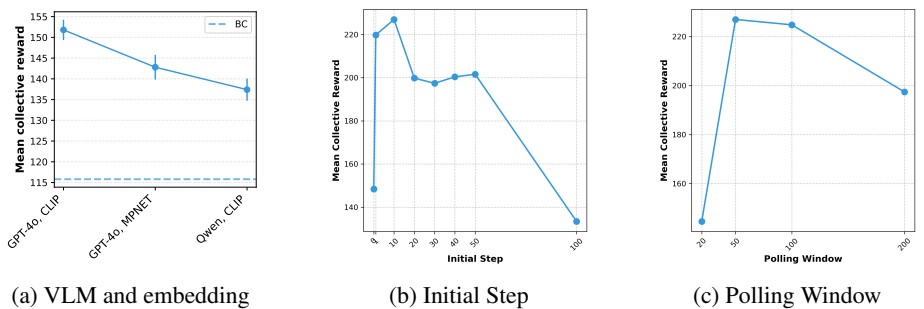

(a) VLM and embedding      (b) Initial Step      (c) Polling Window

Figure 10: Sensitivity on overcooked Cramped-room

### G.5 How efficient is MIMIC during inference?

We confirm the findings of Appendix F empirically by showing in Table 7 that MIMIC's simulation runtime matches that of DDPM-T BC in vision-based environments. Since MIMIC's CVAE is vision-based, it would be unfair to compare against non-vision policy networks.

Table 7: Runtime (s) ↓ for different vision environments.

| Environment | BC | MIMIC |
|---|---|---|
| Aligning | 40.69 | 57.16 |
| Sorting | 71.50 | 78.5 |
| Overcooked | 93.23 | 94.72 |

### G.6 How well does MIMIC enable designer-specified control to generate desired behaviors?

Figure 11 shows examples of different behaviors produced by the MIMIC model conditioned with different descriptions of behaviors. Figure 11a shows a quick repositioning at the start, but due to mediating inner speech, it is not realized later. Figure 11b, on the other hand, shows an attempt to align/match the edges at the start according to the condition. Figure 11c shows an attempt to adjust the box position before pushing straight ahead after a mediation. We find a right side curve approach in Figure 11d, but due to misalignment, it does a lot of rotation at the end. A mediation would have helped here. We find that the zig-zag motion is exhibited in both Figures 11e and 11f while Figure 11f shows more adjustment at the final step as described in the input behavior description.

Table 8: Comparison with other imitation learning models. Here, we use BESO as the base diffusion policy network in MIMIC instead of DDPM-T for fairness.

| Aligning | | | | Overcooked cramped room | |
|---|---|---|---|---|---|
| Model | Success rate | Distance | Entropy | Model | Collective Reward |
| BeT | 0.51667 | 0.12949 | 0.40475 | BeT | $47.2 \pm 4.64$ |
| BeSO | 0.85417 | 0.04954 | 0.6141 | BESO | $67.8 \pm 4.55$ |
| MIMIC-S | **0.88125** | **0.04234** | **0.7215** | MIMIC | $120.2 \pm 2.86$ |
| MIMIC-E | 0.86875 | 0.04759 | 0.7706 | MIMIC-MA | $\mathbf{141.2 \pm 3.68}$ |

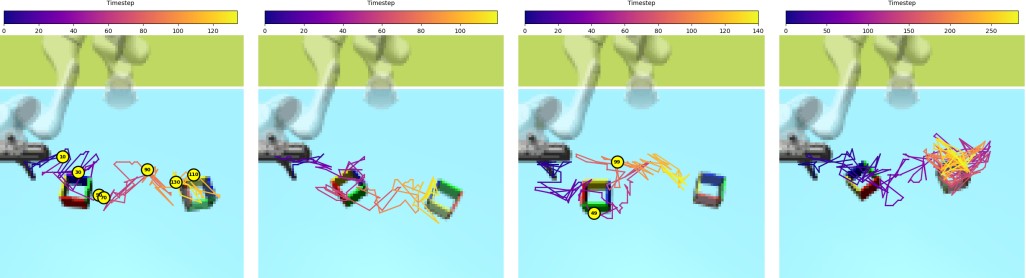

(a) I quickly reposition and push to achieve contact (b) I align the edges first before making full contact. (c) I adjust the box position slightly before making the final approach (d) I curve around from the right side, adjusting for alignment mid-way.

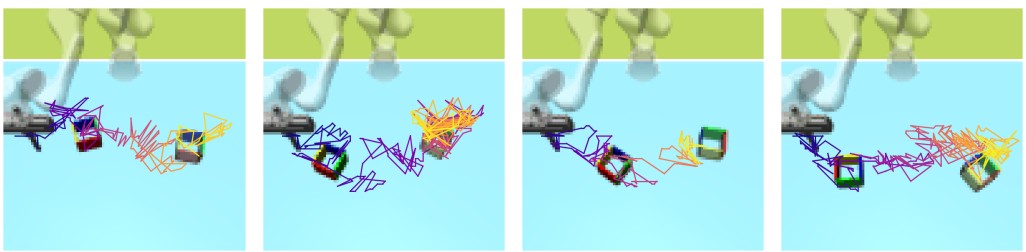

(e) I apply a zigzag path to navigate around an obstacle (f) I use a zigzag pattern to close the distance, adjusting final position carefully (g) I use a swift motion from the top, maintaining a steady path for alignment (h) I begin with a swift side approach, then slow down to ensure precise alignment

Figure 11: Conditional generation on the Aligning dataset. The color gradient (Plasma) shows the simulation time going from 0 (dark purple) to the end (bright yellow), going through red and orange. Inner speech updates are marked as yellow circles with corresponding times inside; the first inner speech is equal to the specified condition.

On the other hand, Figure 11g shows a swift motion moving from the top and fast convergence to the desired box, as mentioned in the input text while Figure 11h begins with a swift approach but then spends a lot of time in the final alignment with the desired box as mentioned in the text, similar to Figure 11f. These results demonstrate significant success achieved by MIMIC towards enabling steerable imitation of desired behaviors.

We also extend this analysis to the Sorting dataset as we find how it prioritizes the closest red block before any blue in Figure 12a, alternate color sorting in Figure 12b, and a behavior of grouping before moving into desired sorted places in Figure 12c.

### G.7 What does generated inner speech look like during simulation?

Since CVAE-generated inner speech lies in the latent embedding space, it is hard to fully interpret and visualize them. We thus employ a heuristic technique to analyze the inner speech during simulation by retrieving the top-2 training descriptions in CLIP's embedding space at each update step. We use the cosine similarity to find the nearest training description and provide the value in parentheses.

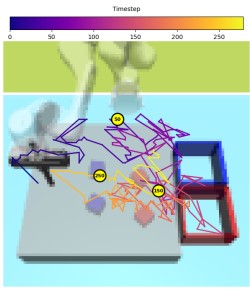

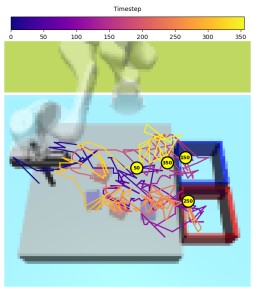

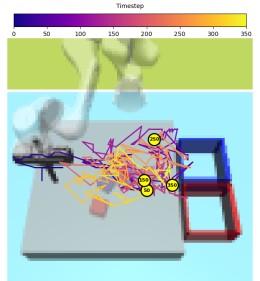

(a) I prioritize the closest red block before any blue.

(b) I alternate between picking a red block and a blue block.

(c) I sort blocks by color, grouping similar colors together before moving them.

Figure 12: Conditional generation on the Sorting dataset. The color gradient (Plasma) shows the simulation time going from 0 (dark purple) to the end (bright yellow) going through red and orange. Inner speech updates are marked as yellow circles with corresponding times inside; the first inner speech is equal to the specified condition.

Here, we provide examples for both conditional and unconditional simulations and find how the top captions change along with the similarity score.

### G.7.1 Conditional

**I rotate the box first before aligning it with the target.**

| Timestep | Closest description | Similarity |
|---|---|---|
| t=49 | I push the box directly without rotation, aiming for a straightforward alignment. | 0.9183 |
| t=99 | I approach directly, then rotate in place to align perfectly. | 0.9614 |
| t=149 | I approach directly, then rotate in place to align perfectly. | 0.9614 |
| t=249 | Starting from a slightly rotated angle, I need a mid-action adjustment to align. | 0.9616 |

**I begin with a swift side approach, then slow down to ensure precise alignment.**

| Timestep | Closest description | Similarity |
|---|---|---|
| t=49 | I approach the box from the side, rotating slightly to align smoothly with the fixed box. | 0.9471 |
| t=99 | I approach directly, then rotate in place to align perfectly. | 0.9614 |
| t=149 | I approach directly, then rotate in place to align perfectly. | 0.9619 |
| t=249 | I approach with a direct path but make a last-second adjustment to align perfectly. | 0.9611 |

**I use a swift motion from the top, maintaining a steady path for alignment.**

| Timestep | Closest description | Similarity |
|---|---|---|
| t=49 | I execute a direct push from behind, minimizing lateral movement. | 0.9477 |
| t=99 | I carefully approach from the left, ensuring alignment from a diagonal perspective. | 0.9613 |
| t=149 | Starting from a slightly rotated angle, I need a mid-action adjustment to align. | 0.9613 |
| t=249 | I approach with a direct path but make a last-second adjustment to align perfectly. | 0.9613 |

### G.7.2 Unconditional

**Aligning**

| Timestep | Closest description | Similarity |
|---|---|---|
| t=0 | I start with a straight approach from a central position, requiring minimal rotation for alignment. | 0.9693 |
| t=50 | I start with a straight approach from a central position, requiring minimal rotation for alignment. | 0.9691 |
| t=100 | I start with a straight approach from a central position, requiring minimal rotation for alignment. | 0.9696 |

**Sorting**

| Timestep | Closest description | Similarity |
|---|---|---|
| t=100 | I focus on sorting all blocks of one color first before switching to the other. | 0.9607 |
| t=300 | I focus on sorting all blocks of one color first before switching to the other. | 0.9604 |

**Overcooked Cramped room**

| Timestep | Closest description | Similarity |
|---|---|---|
| t=100 | I quickly grab onions from the pile and place them in the pot, prioritizing speed over precision. | 0.9082 |
| t=200 | I quickly grab onions from the pile and place them in the pot, prioritizing speed over precision. | 0.9096 |
| t=300 | I quickly grab onions from the pile and place them in the pot, prioritizing speed over precision. | 0.9062 |
| t=400 | I quickly grab onions from the pile and place them in the pot, prioritizing speed over precision. | 0.9086 |

**Overcooked Coordination ring**

| Timestep | Closest description | Similarity |
|---|---|---|
| t=60 | I adjust its movement pattern to avoid congestion, optimizing its task execution efficiency. | 0.9521 |
| t=110 | I adjust its movement pattern to avoid congestion, optimizing its task execution efficiency. | 0.9405 |
| t=160 | I adjust its movement pattern to avoid congestion, optimizing its task execution efficiency. | 0.9466 |
| t=210 | I adjust its movement pattern to avoid congestion, optimizing its task execution efficiency. | 0.9407 |
| t=260 | I adjust its movement pattern to avoid congestion, optimizing its task execution efficiency. | 0.9395 |

**Overcooked Asymmetric Advantages**

| Timestep | Closest description | Similarity |
|---|---|---|
| t=50 | I maneuver around obstacles effectively. | 0.8391 |
| t=100 | I balance interactions with other agents and time efficiency. | 0.8416 |
| t=150 | I balance interactions with other agents and time efficiency. | 0.8463 |
| t=200 | I maneuver around obstacles effectively. | 0.8443 |
| t=250 | I balance interactions with other agents and time efficiency. | 0.8401 |

