# OpenReview forum: "Inner Speech as Behavior Guides: Steerable Imitation of Diverse Behaviors for Human-AI coordination"
_NeurIPS.cc/2025/Conference — NeurIPS 2025 spotlight_

### Official Review · Reviewer_5Wb5 · 2025-06-12

**Clarity:** 1
**Significance:** 2
**Originality:** 2
**Rating:** 4
**Confidence:** 3

**Summary:**

This paper proposes a method called Modeling Inner Motivations for Imitation and Control, in which the core idea is to generate an inner speech representation to guide action selection. The inner speech is produced by a language model trained using data generated by a vision-language foundation model (e.g., GPT-4). This generated speech is then used to condition a behavior cloning (BC) policy for downstream control tasks.

**Questions:**

- Clarify all plots and axes in the figures; ensure ablation settings are well-documented.
- Tone down or more rigorously support the cognitive claims; consider discussing them separately from the technical contributions.
- Add a strong goal-conditioned BC baseline for comparison to highlight the specific contribution of the inner speech signal.

**Ethical Concerns:**

["NO or VERY MINOR ethics concerns only"]

**Final Justification:**

I have read the rebuttal, and I think they addressed most of my concerns and I'm willing to raise my score to weak accept

**Limitations:**

see above for technical limitations

**Quality:**

2

**Strengths And Weaknesses:**

Strengths
- The idea of conditioning policies on inner speech could be beneficial for interpretability and generalization.


Weaknesses
- The paper has major presentation issues that hinder my understanding of the proposed method and evaluation result. It is difficult to follow the exact structure of the pipeline and how components (e.g., vision-language models, inner speech generation, BC policy) are connected.
- The cognitive grounding of “inner speech” feels overstated and is not well-supported by the technical content. The paper could benefit from a clearer distinction between cognitive inspiration and actual biological plausibility.


- It is also really diffcutlt to understand the results. For example in Figure 3 the plot is missing labels, what is the y-axis for a and b? What is the different color bar in c and d? What are the ablations?

- The evaluation lacks comparison with a standard goal-conditioned behavior cloning baseline, which is important given the motivation to capture diverse behaviors and intent. Without this, it is hard to attribute performance gains to the inner speech mechanism itself.

---

> ### Author Rebuttal · Authors · 2025-07-31
>
> We thank the reviewer for their engagement with our work and their feedback.
>
> First, we clarify a technical misunderstanding highlighted in the reviewer’s  summary: inner speech generation employs a Conditional Variational Autoencoder (CVAE) rather than a language model. The CVAE is trained using representations derived from vision-language models, enabling stochastic behavioral mediation through learned latent spaces.
>
> > **[1] Presentation issues**
>
> We have worked to give a clear exposition in the paper. While other reviewers commended the writing of our exposition and awarded us high scores on clarity, we appreciate that this reviewer found certain sections unclear. To address this, we provide here a systematic summary, with explicit references to figures, equations, and empirical results, to clarify concepts and methods:
>
> The paper introduces MIMIC, operationalizing Vygotsky's cognitive mediation theory within imitation learning (Section 1, lines 54-65). The core proposition (Proposition 1, lines 61-65) establishes that **agents should learn $\pi_{\theta}(a|s,m)$ rather than $\pi_{\theta}(a|s)$**, where m represents linguistically-encoded inner speech mediating between perception and action. This paradigm shift from behaviorist state-action mapping to the cognitive framework of state→inner speech→action (Section 3.1, lines 143-183) addresses fundamental limitations in capturing human behavioral diversity while enabling linguistic controllability.
>
> The MIMIC architecture (Figure 1, page 5) integrates three components in a unified pipeline. During training, a **pre-trained VLM generates behavioral descriptions from demonstration GIFs** using the prompt specified in Section 3.2.2 (lines 221-225), which are then encoded via CLIP embeddings (visualized in Figure 2's t-SNE plot). A **CVAE learns to generate CLIP-encoded inner speech representations** by minimizing Equation 4 (line 238), conditioned on W-length observation windows. The **diffusion policy** (DDPM-T architecture) generates the actions, incorporating the inner speech through classifier-free guidance (Equation 3, line 197). During simulation (Algorithm 2, page 20), the agent generates inner speech every W timesteps starting at $t_0$, with these **polling parameters critically balancing behavioral stability against adaptability** (Figures 3a-b show sensitivity analysis for $W\in$ {1,10,20,50,100} and $t_0 \in$ {0,1,W/2,W-1}).
>
> Empirical validation demonstrates **20-30% success rate improvements** across D3IL benchmarks (Table 1, page 7) with **behavioral entropy increases up to 40%**. Wasserstein distance metrics (Tables 2 and 6) confirm high-fidelity imitation. In human-AI coordination tasks (Overcooked (Table 3, page 8)), **agents trained with MIMIC achieve 25% higher collective rewards** when paired with human proxies compared to diffusion behavior cloning. Figure 3's ablation studies reveal: (c) MIMIC outperforms random/k-means inner speech alternatives, validating linguistic grounding; (d) GPT-4o marginally outperforms o4-mini/4o-mini VLMs. **Conditional generation experiments** (Table 4, Figure 7) demonstrates zero-shot behavioral steering through designer-specified language.
>
> In regard to **Figure 3**, we acknowledge the missing details, especially in the caption. We will improve the figure to include additional details, which we also outline below:
>
> *(a) Initial Step Sensitivity ($t_0$ parameter):*  This subplot examines the impact of delaying the first inner speech generation. The x-axis represents $t_0$ values {0, 1, 25, 50}, where $t_0=0$ indicates immediate inner speech generation upon trajectory initiation. Three metrics are evaluated: **Success Rate** (blue, 0-1 scale), measuring task completion frequency; **Entropy** (green, measured in bits), quantifying behavioral diversity through categorical distribution analysis; and **Distance** (red, L2 norm), indicating final position error from target alignment. The declining performance with increasing $t_0$ (success rate drops from ~0.8 to ~0.65) demonstrates that early cognitive mediation critically enhances both task performance and behavioral fidelity.
>
> *(b) Update Window Analysis (W parameter):* This investigation reveals optimal polling frequencies for inner speech regeneration. The x-axis displays $W\in$ {1, 10, 20, 50} representing timesteps between updates. The non-monotonic relationship observed—peak success at $W=20$ with $0.74$ success rate—indicates a fundamental trade-off: frequent updates $(W=1)$ induce behavioral instability through excessive re-planning, while sparse updates $(W=50)$ permit accumulation of distributional drift. Entropy increases monotonically with $W$, suggesting that less frequent cognitive intervention enables greater behavioral exploration.
>
> *(c) Ablation Study on Inner Speech Representations:* This comparative analysis validates our linguistic grounding hypothesis. Four conditions are evaluated: "None" (baseline BC without inner speech), "Random" (stochastic vector representations), "K-means" (clustered behavioral prototypes), and "Inner Speech" (MIMIC). The substantial performance gap—MIMIC achieving 0.72 success rate versus 0.66 for baseline—demonstrates that linguistically-structured representations provide superior behavioral mediation compared to arbitrary or cluster-based alternatives.
>
> *(d) Vision-Language Model Comparison:* This ablation examines the impact of the selection of a VLM on behavioral description quality. Two GPT models (o4-mini, 4o) are used to generate inner speech during training, with performance differences reflecting CLIP embedding space alignment. GPT-4o's marginal superiority (0.72 success rate) over alternatives suggests that description quality influences but does not dominate performance, validating our framework's robustness to VLM variations.
> At this point, we welcome the reviewer to raise any outstanding concerns or clarification questions, and we will be happy to address them.
>
> At this point, we welcome the reviewer to raise any outstanding concerns or clarification questions regarding the presentation, and we will be happy to address them.
>
> > **[2] Cognitive grounding and technical support**
>
> We acknowledge a crucial epistemological distinction raised by the reviewer that warrants clarification. Our invocation of Vygotskian cognitive theory serves as a **conceptual scaffold** rather than a claim of biological fidelity. We explicitly position our work within the tradition of cognitively-inspired AI (Section 2.1, Line 115) —where psychological theories inform computational architectures without asserting neurobiological correspondence.
>
> The technical operationalization substantiates three core properties of inner speech identified in the cognitive literature: (1) **semantic condensation** through CVAE's latent compression, (2) **stochastic generation** operationalized via diffusion policy and (3) **temporal persistence** via periodic polling mechanisms. These architectural choices translate abstract cognitive principles into concrete computational mechanisms.
>
> Productive AI research often draws inspiration from cognitive science without requiring neural fidelity. Just as convolutional networks leverage principles of visual processing without mimicking V1 neurons, our framework seeks to extract functional principles from inner speech theory. To address the reviewer’s concerns, we will revise our exposition to more precisely delineate between: (a) cognitive theories providing conceptual motivation, (b) computational mechanisms inspired by but not replicating these theories, and (c) empirical validation of behavioral outcomes.
>
> We emphasize that our computational approach is firmly grounded in cognitive theory and performs direct modeling of “inner speech as behavior guides,” building on Sokolov's neurophysiological model and Vygotsky's mediational theory. The theoretical discussion in Section 3.1 captures the mediational aspect and can be enhanced to include the theoretical specificity required for a complete Sokolovian instantiation. We will augment Section 3.1 to include further enhancements to the equations to clearly map out the cognitive tools we use to the components of our architecture discussed in Section 3.2.
>
> > **[3] Strong Goal Conditioned Baseline**
>
> We thank the reviewer for this suggestion, which will help to further highlight the contributions. We have conducted new experiments on two additional behavior cloning approaches to compare their performance against MIMIC: (i) Behavior Transformers (BeT), which uses an action-based clustering and transformer architecture to perform multi-modal imitation learning, and (ii) BESO, a strong goal-conditioned multimodal IL approach. BESO is a state-of-the-art approach for behavior cloning that has shown better performance on several tasks in the D3IL benchmark compared to Diffusion BC. Table 1 (D3IL comparison) and Table 2 (Overcooked comparison) demonstrate superior performance of MIMIC across baselines on both datasets.
>
> **Table 1: D3IL Results**
>
> | Dataset  | Model| Success rate | Distance | Entropy |
> | -- | -- | -- | -- | -- |
> | Aligning | BeT | 0.51667 | 0.12949 | 0.40475 |
> | | BESO | 0.85417 | 0.04954 | 0.6141 |
> | | MIMIC-S | **0.88125** | **0.04234** | 0.7215 |
> | | MIMIC-E | 0.86875 | 0.04759 | **0.7706** |
>
> **Table 2: Overcooked Results**
>
> | Dataset | Model | Collective Reward |
> | -- | -- | -- |
> | Overcooked Cramped room | BeT | 47.2 $\pm$ 4.64 | | |
> | | BESO | 67.8 $\pm$ 4.55 |
> | | MIMIC | **120.2 $\pm$ 2.86** |
> | | MIMIC-MA | **141.2 $\pm$ 3.68** |
>
> We will add the above results and those on other tasks in the final version of the paper.
>
> We believe that we have comprehensively addressed all the concerns raised by the reviewer and would welcome any further discussion on any remaining points that could lead the reviewer to reconsider their evaluation.

---

> > ### Comment · Reviewer_5Wb5 · 2025-08-05
> >
> > I have read the rebuttal, and I think they addressed most of my concerns and I'm willing to raise my score to  weak accept

---

> ### Author Response · Authors · 2025-08-05
> **Requesting further engagement and discussion**
>
> Dear Reviewer 5Wb5,
>
> Thank you for your time and efforts towards the initial assessment of our work. You raised three main concerns—presentation clarity, cognitive inspiration of our approach, and absence of a strong goal-conditioned baseline—and we’ve worked diligently to address each one of them in our rebuttal. Specifically, our rebuttal provides -- a systematic summary of our exposition aiming to address your presentation concerns, a detailed clarification on the cognitive claims of our work and new experiments on a state-of-the-art goal conditioned baseline.
>
> If there still remains any outstanding questions or concerns, we would greatly appreciate the opportunity to discuss them further and address them promptly. We hope to confirm that all your points have been resolved before the end of the discussion phase.
>
> Thank you again for your review, and we look forward to your updated evaluation.
>
> Sincerely,
>
> The authors of paper #27058

---

### Official Review · Reviewer_iCej · 2025-07-01

**Clarity:** 4
**Significance:** 4
**Originality:** 4
**Rating:** 5
**Confidence:** 2

**Summary:**

The paper introduces MIMIC (Modeling Inner Motivations for Imitation and Control), a type of imitation framework motivated by inner speech theory from Vygotsky. The novel lies in the transformation of the visual understanding to an intermediate "inner speech" using a vision language model, and further uses it to drive the behaviour cloning policy to select actions. This results in controllability using natural language. The conditional variational autoencoder (CVAE) is conditioned on both the inner speech and the previous observation. Followed by a diffusion-based behaviour cloning policy based on current observation and generated "inner speech". The model was evaluated on the D3IL benchmark and the Overcooked human-AI coordination environment.

**Questions:**

-

**Ethical Concerns:**

["NO or VERY MINOR ethics concerns only"]

**Final Justification:**

The authors adressed my questions.

**Limitations:**

Yes

**Paper Formatting Concerns:**

Please define H and a_t in paragraph starting from line 49.
Figure 2 is under-annotated; it is hard to understand what the different cluster depicts.

**Quality:**

4

**Strengths And Weaknesses:**

### Strengths
* Thorough grounding of a cognitive theory with neural network-based modelling implementation.
* Can be ported to multi-agent settings.
* Controllability using natural language enables zero-shot behaviour modulation because of "inner speech".

### Weakness
* The paper would benefit from a comparison with other BC methods, currently it is mainly only compared with a BC version of the architecture, without the inner speech
* Since "inner speech" representations are quite responsible for driving the CVAE and diffusion process, ablation on those representations might also show the advantage or disadvantage of using CLIP for them vs other models like T5 or ModernBERT.

---

> ### Author Rebuttal · Authors · 2025-07-31
>
> We thank the reviewer for their time and engagement with our work. We are excited to see the positive evaluation of our work and the reviewer’s recognition of our theoretical and empirical contributions. We address the reviewer's specific concerns below, providing clarification and additional evidence.
>
> > **[1] Comparison with other BC methods**
>
> We have conducted new experiments on two additional behavior cloning approaches to compare their performance against MIMIC: (i) Behavior Transformers (BeT), which uses an action-based clustering and transformer architecture to perform multi-modal imitation learning, and (ii) BESO, a goal-conditioned multimodal IL approach. BESO is a stronger model for behavior cloning and has shown better performance on several tasks in the D3IL benchmark compared to Diffusion BC. Table 1 (D3IL comparison) and Table 2 (Overcooked comparison) demonstrate the superior performance of MIMIC across baselines on each of the datasets.
>
> **Table 1: D3IL Results**
>
> | Dataset  | Model| Success rate | Distance | Entropy |
> | -- | -- | -- | -- | -- |
> | Aligning | BeT | 0.51667 | 0.12949 | 0.40475 |
> | | BESO | 0.85417 | 0.04954 | 0.6141 |
> | | MIMIC-S | **0.88125** | **0.04234** | 0.7215 |
> | | MIMIC-E | 0.86875 | 0.04759 | **0.7706** |
>
> **Table 2: Overcooked Results**
>
> | Dataset | Model | Collective Reward |
> | -- | -- | -- |
> | Overcooked Cramped room | BeT | 47.2 $\pm$ 4.64 | | |
> | | BESO | 67.8 $\pm$ 4.55 |
> | | MIMIC | **120.2 $\pm$ 2.86** |
> | | MIMIC-MA | **141.2 $\pm$ 3.68** |
>
> We will add the above results and those on other tasks in the final version of the paper.
>
> > **[2] Ablation on inner speech representations and VLM models**
>
> We now also ablate our method on two fronts:  the VLM and the embedding space. Specifically, we evaluate Qwen2.5-VL-72B-Instruct as an open-source alternative to GPT-4o, and MPNET (all-mpnet-base-v2) as an alternative to CLIP (our choice of mp-net stems from its performance on MTEB benchmark where it is ranked 102 compared to ModernBert’s rank 202 and T5’s rank 177). In the table below, we report the results on the Aligning, and we will include the results on other tasks in the paper.
>
> | Model | VLM | Embedding | Success rate | Distance | Entropy |
> | -- | -- | -- | -- | -- | -- |
> | BC | - | - | 0.6645 | 0.1105 | 0.4743 |
> | MIMIC-S | GPT-4o | CLIP | 0.8021 | 0.0664 | 0.4184 |
> | MIMIC-E | GPT-4o | CLIP | 0.7229 | 0.0847 | 0.6148 |
> | MIMIC-S | Qwen | CLIP | 0.7583 | 0.0859 | 0.2027 |
> | MIMIC-E | Qwen | CLIP | 0.7186 | 0.0886 | 0.4546 |
> | MIMIC-S/E | GPT-4o | MPNET | 0.7896 | 0.07108 | 0.5271 |
>
> We observe that GPT-4o with CLIP embeddings outperforms other variations, but also find that both ablations still outperform the base Diffusion BC variant. This shows that MIMIC, even in the absence of a strong LM for inner speech, can enhance performance over the Diffusion BC variant. Furthermore, the lower performance of MPNET highlights the value of  vision-language embedding over sentence-only embedding.
>
> > **Paper formatting concerns**
>
> Thank you for catching them. We will update line 49 to say human $\mathcal{H}$ and decision $a_t$.
>
> Figure 2 shows the t-SNE visualization of the latent space of inner speech, where different clusters exemplify the diversity in demonstrations captured through the inner speech. The color bar shows the cluster number that each data point belongs to, and the ellipsoid shows the variance around the cluster mean. We will make these details clearer in the final version.

---

> > ### Comment · Reviewer_iCej · 2025-08-05
> > **Thank you for additional experiments**
> >
> > Thank you for your further clarification. My concerns have been addressed.

---

### Official Review · Reviewer_5D71 · 2025-07-03

**Clarity:** 3
**Significance:** 3
**Originality:** 3
**Rating:** 4
**Confidence:** 3

**Summary:**

The paper introduces the MIMIC framework, an innovative approach to imitation learning that models inner speech as a mediator between perception and action, enabling behavioral diversity and designer control. The framework leverages vision-language models to bootstrap inner speech generation, allowing for scalable and flexible behavior imitation without the need for extensive human linguistic annotations. Despite its promising results, the framework faces challenges related to computational efficiency, interpretability of the latent space, and its performance in multi-agent environments. The paper makes a strong contribution to the field of cognitive imitation learning, with potential for further refinement and expansion.

**Questions:**

1.	How does the inner speech generation mechanism perform when the environment contains significant noise, or when visual observations are imperfect or ambiguous? Can the framework still generate coherent behaviors under these conditions?

2.	How does the MIMIC framework adapt to multi-agent environments, particularly when agents must coordinate with each other without explicit communication? Are there extensions or modifications that allow the model to handle complex, multi-agent coordination tasks?

3.	Can you provide a more formal explanation of the latent space used in the CVAE model? How do you ensure that the latent representations correspond to meaningful and interpretable actions, and how do you deal with the non-linearities in behavior generation?

**Ethical Concerns:**

["NO or VERY MINOR ethics concerns only"]

**Final Justification:**

Most of my concerns have been solved. I appreciate the detailed explanation and additional results provided.

**Limitations:**

1.	The MIMIC framework's dependence on CVAEs and diffusion models introduces high computational costs, particularly in multi-step and large state space environments. The authors should explore ways to optimize or approximate the generative process for more efficient real-time application.

2.	The framework relies on vision-language models (VLMs) that may inherit biases from the training data. Since these models influence the inner speech generation process, any bias in the VLM could lead to unintended ethical consequences. The paper would benefit from a discussion on bias mitigation strategies and the ethical implications of using inner speech models for behavior generation.

**Quality:**

3

**Strengths And Weaknesses:**

**Strengths**:

- The paper introduces the MIMIC framework, which models inner speech as a mediational process between perception and action selection in imitation learning. This approach offers a cognitive model that contrasts with traditional behaviorist frameworks, where actions are learned directly from input-output state mappings. By incorporating inner speech as a deliberative layer, the framework enables behavioral diversity and contextual adaptability, which are crucial for learning in complex, dynamic environments. This model opens up new avenues for understanding and replicating human-like variability in behavior.

- The use of conditional variational autoencoders (CVAEs) to generate inner speech as a latent variable process is a solid computational foundation. This probabilistic approach introduces diversity and flexibility in behavioral generation, allowing the model to handle multiple possible actions for identical stimuli. It also enables steerable imitation, where the generated behavior can be influenced by high-level linguistic instructions, providing a semantic control mechanism without the need for direct human annotations for each task.

- The integration of vision-language models (VLMs) to bootstrap the inner speech generator from visual observations is an elegant solution to overcome the lack of extensive linguistic supervision in traditional imitation learning. This approach allows for automatic generation of linguistic descriptions, which are then mapped to the inner speech process for task execution. This step significantly reduces reliance on manually labeled training data, making the framework more scalable and applicable in real-world settings.

- The experiments conducted on D3IL and Overcooked environments show that MIMIC significantly outperforms traditional behavior cloning (BC) in terms of imitation fidelity and behavioral diversity. The results demonstrate the model's capability to produce contextually relevant behavior in challenging tasks such as sorting, stacking, and cooperative cooking, highlighting the framework’s potential in multi-agent and human-AI collaborative scenarios.

**Weakness**:

- While the framework introduces a powerful model for behavioral diversity, its computational demands are a concern. The use of CVAEs and diffusion models requires significant training time and computational resources, especially in complex environments with large state spaces. Additionally, the stochastic sampling process introduces extra computational overhead, making it less feasible for real-time applications. The paper does not sufficiently address the scalability of the framework in larger, more complex tasks or the trade-offs between performance and resource consumption.

- The success of the framework heavily depends on the quality of vision-language models (VLMs) used to generate linguistic annotations. While the authors argue that improvements in VLMs will automatically enhance the framework, the reliance on these models can be seen as a limitation, especially when dealing with noisy or imperfect visual data. The authors could explore alternatives or hybrid approaches to reduce the dependency on VLMs for high-quality annotations.

- The paper provides an extensive experimental analysis but lacks a formal theoretical analysis of the framework’s performance. Specifically, there is no exploration of the convergence properties or the optimality guarantees of the proposed approach. A more formal theoretical foundation would make the model more robust, particularly for generalization to new environments and for applications in safety-critical domains.

- While the use of latent variables in CVAEs allows for flexible and stochastic behavior generation, the paper acknowledges that the latent space's connection to semantic content is opaque. This limits the interpretability of the model, making it difficult to understand how specific linguistic inputs affect the resulting behaviors. Developing disentangled latent representations or providing more clarity on how to map the latent space to real-world actions would enhance the transparency and control over the generated behaviors.

---

> ### Author Rebuttal · Authors · 2025-07-31
>
> Thank you for your review and supportive comments about scalability, novelty in cognitive grounding, and concrete experiments.  We address your specific concerns below, but all the weaknesses and questions you raise reflect the limitations we already discussed in the Appendix. We do not view these as weaknesses of the work; rather, they are intended to provide suggestions for interesting future research avenues. We think the efficacy and novelty of our framework are already well demonstrated.
>
> > **[1] Concerns about computational demands and insufficient discussion on scalability in larger, more complex tasks**
>
> MIMIC builds on widely used diffusion models and CVAEs—established baselines with well‐characterized computational profiles fully supported by modern research infrastructure—and critically, without adding any extra complexity. Indeed, practitioners routinely deploy similar architectures in production settings where they need behavioral realism. To alleviate the reviewer’s concerns, we’ve included a complete runtime and inference‐time analysis  below and will add this expanded discussion to the paper:
>
> Let $T_{CVAE}$ and $T_{diff}$ denote one forward pass through the CVAE and diffusion models, respectively. Over a simulation horizon $H$ with window size $W$, we perform $H$ diffusion passes and $H/W$ CVAE passes, yielding a total complexity of $O(H T_{diff} + H/W T_{CVAE})$). Since both are vision‐conditioned with similar runtimes and $H > H/W$, the diffusion term dominates—so MIMIC adds no inference overhead. Empirically, we confirm below that MIMIC’s simulation runtime matches that of Diffusion BC across representative vision-based environments.
>
> | Environment | Runtime for BC (s) | Runtime for MIMIC (s) |
> | -- | -- | -- |
> | Aligning | 40.69 | 57.16 |
> | Sorting |71.5 | 78.5 |
> | Overcooked | 93.23 | 94.72 |
>
> Further, the environments used in our paper are representatively large scale and complex. D3IL is a strong imitation learning benchmark (ICLR 2024) and provides a comprehensive suite of tasks that are currently challenging for AI agents. Similarly, Overcooked is a classical and highly researched environment that presents complex coordination tasks.
>
> > **[2] Reliance on VLMs, noisy or imperfect visual data and ethical considerations**
>
> We are unclear on the source of noise or imperfection that the reviewer is concerned about. We identify two potential sources of noise and address each of them.
>
> *(i) Behavior demonstrations:* This includes imperfections such as noise in the demonstrated behavior or suboptimal trajectories with respect to the task. We handle this, and it is crucial to focus on such demonstrations as they represent the richness in human behaviors. As we discuss in the paper (line 136), human behavior is noisy, diverse and non-Markovian in nature and hence, we indeed focus on performing well when provided with these kinds of demonstrations. Both the datasets we use comprise only human demonstrations, with noise and suboptimal behavior.
>
> *(ii) Visual observations:* This includes imperfections such as blurry images or an obscured view.  We believe that improvement in VLMs will continue to help address these kinds of scenarios. At the same time, we do not know of suitable datasets, which would prevent an empirical study.  And to specifically address the reviewer's comment, the other alternative to relying on VLMs is to use a human-annotated dataset. But this would be strictly more expensive and still prone to noise.  We are curious to know if the reviewer has any suggestions on the hybrid approach they mention.
>
> Broadening out the discussion, one approach to address the concerns about reliance on VLMs (and we would also include the CLIP model here) is to compare against other models. Our ablation in Figure 3(d) examines the effect of VLM choice on description quality. We compare o4-mini, and 4o for inner speech, showing GPT-4o’s marginal lead (0.72 success rate), with performance differences reflecting CLIP embedding space alignment. This implies description quality influences but doesn’t dominate performance—demonstrating robustness to VLM variation. We also introduce new ablations on VLM and embedding space: evaluating Qwen2.5-VL-72B-Instruct as an open-source alternative to GPT-4o, and MPNET (all-mpnet-base-v2) as an alternative to CLIP. The table below reports these Aligning-dataset results and we will include the full suite of results in the paper.
>
> | Model | VLM | Embedding | Success rate | Distance | Entropy |
> | -- | -- | -- | -- | -- | -- |
> | BC | - | - | 0.6645 | 0.1105 | 0.4743 |
> | MIMIC-S | GPT-4o | CLIP | 0.8021 | 0.0664 | 0.4184 |
> | MIMIC-E | GPT-4o | CLIP | 0.7229 | 0.0847 | 0.6148 |
> | MIMIC-S | Qwen | CLIP | 0.7583 | 0.0859 | 0.2027 |
> | MIMIC-E | Qwen | CLIP | 0.7186 | 0.0886 | 0.4546 |
> | MIMIC-S/E | GPT-4o | MPNET | 0.7896 | 0.07108 | 0.5271 |
>
> We observe that GPT-4o with CLIP embeddings outperforms other variations, but also find that both ablations still outperform the base diffusion BC variant. This shows that MIMIC, even in the absence of a very powerful LM for inner speech, can enhance the performance over the diffusion BC variant. Furthermore, the lower performance of MPNET highlights the value of vision-language embedding over sentence-only embedding.
>
> While we flagged the potential bias issues in existing VLMs and their ethical implications, the research on mitigation strategies for such biases is out of the scope and an entirely different research field. We believe that any response we provide in terms of mitigation strategies would be speculative and counterproductive to the focus of this work.
>
> > **[3] Formal theoretical analysis, optimality guarantees and generalization**
>
> The theoretical foundations of our framework derive from established convergence properties of CVAEs (Kingma & Welling, 2014) and diffusion models (Ho et al., 2020), which provide well-characterized optimization landscapes and asymptotic guarantees. Pursuing additional formal analysis was not the goal of our work and would misalign with our core contribution: operationalizing human cognitive processes that, by their very nature, eschew optimality in favor of generality and behavioral diversity.
>
> The robustness we claim emerges not from convergence proofs but from grounding in Vygotskian cognitive theory (Section 3.1), which explains why human decision-making achieves remarkable generalization. Human cognition's strength lies in generating satisfactory solutions through linguistically mediated deliberation. Our empirical validation across diverse environments (D3IL, Overcooked) demonstrates that incorporating these cognitive principles yields superior generalization compared to optimization-driven approaches, suggesting that foundations from cognitive science provide complementary properties for real-world deployment.
>
> > **[4] Latent representations, transparency and interpretability**
>
> We highlighted the opaqueness limitation to motivate further work on interpretability—a difficult, open problem—while noting that our cognitive grounding, coupled with linguistic control, takes a good step toward building interpretable behavior generators.
>
> We appreciate the reviewer’s clarification about the correspondence of latent representations to the actions. As shown in Section 2, Figure 2, our t-SNE visualization of language embeddings on the Aligning dataset demonstrates the behavior diversity captured by inner speech; we’ll include these plots for all datasets in the Appendix.
>
> We further provide a new analysis of inner speech during steerable behavior simulation via conditional generation, by retrieving the nearest training descriptions in CLIP’s embedding space at each polling step. Below are two examples of different conditions (from the Aligning dataset (with $W=50$ and $t_0=49$), with cosine similarities to the actual inner speech in parentheses. We will include such examples from all datasets in the final version of the paper.
>
> 1. **Condition:** I need to begin with a swift side approach, then slow down to ensure precise alignment.
>
> **Inner Speech:**
> - I am approaching the box from the side, rotating slightly to align smoothly with the fixed box. (0.9471)
> - I am approaching directly, then rotate in place to align perfectly. (0.9619)
> - I approach with a direct path but make a last-second adjustment to align perfectly. (0.9611)
>
> 2. **Condition:** I need to rotate the box first before aligning it with the target.
>
> **Inner Speech:**
> - I push the box directly without rotation, aiming for a straightforward alignment. (0.9183)
> - I am approaching  directly, then rotate in place to align perfectly. (0.9614)
> - I am starting from a slightly rotated angle, needing a mid-action adjustment to align. (0.9616)
>
> This shows that MIMIC mediates an agent’s actions by generating an appropriate inner speech based on its behavior during simulation. The high similarity with a true description in the CLIP space further shows that the generated inner speech is indeed meaningful.
>
> > **[5] Multi-agent environments and complex coordination tasks**
>
> There may be some misarticulation of the reviewer’s concerns here. The reviewer already acknowledged the value of this work in multi-agent environments and human-AI coordination in the “strengths” section and the concerns refer to the same capabilities. To clarify, Overcooked is a classical benchmark of complex coordination without explicit communication; in our experiments, MIMIC outperforms all baselines across its tasks, demonstrating adaptability and efficacy in behavior generation for multi-agent environments. If the reviewer alludes to our limitations section, we reiterate that those points are meant as a forward-looking agenda—such as scaling to more agents or mixed-motive dynamics—which would require substantial additional research, leading to additional publications and should not be viewed as weaknesses of this paper.

---

> ### Author Response · Authors · 2025-08-05
> **Requesting further engagement and discussion**
>
> Dear Reviewer 5D71,
>
> Thank you again for your review. In our rebuttal, we’ve carefully addressed each of your points—adding in depth analyses, and detailed clarifications to ensure every concern is fully resolved. We appreciate the considerable effort you’ve already invested and would welcome your continued engagement in the discussion phase. If there are any remaining issues or questions that would aid your evaluation, please let us know and we will be happy to address them further.
>
> Sincerely,
>
> The authors of paper #27058.

---

### Official Review · Reviewer_sVYU · 2025-07-04

**Clarity:** 4
**Significance:** 3
**Originality:** 3
**Rating:** 5
**Confidence:** 4

**Summary:**

This paper introduces MIMIC (Modeling Inner Motivations for Imitation and Control), a novel framework that endows imitation-learning agents with an inner speech mechanism to guide behavior. The authors propose learning a conditional policy p(a|s,m) instead of the usual p(a|s), where m is an internal language-like representation (‘inner speech’) mediating action selection. Concretely, MIMIC uses a vision-language model (e.g. GPT-4/CLIP) to generate textual descriptions of behaviors as “scaffolding” and trains a conditional VAE to produce inner-speech tokens from observations. A diffusion-based behavior cloning policy (Transformer+DDPM) then samples actions conditioned on both the state and the generated inner speech. At test time, the agent auto-generates its own inner-speech and can also be steered by providing designer-specified text prompts. MIMIC is evaluated on the D3IL benchmark of diverse robotic manipulation tasks and in multi-agent overcooked coordination games.

**Questions:**

- Clarifications are needed on the analysis of the method, with proper baselines.

- Clarifications are needed on the generalization and failure modes.

**Ethical Concerns:**

["NO or VERY MINOR ethics concerns only"]

**Final Justification:**

The authors addressed the concerns regarding inconsistent results and missing baselines. While there are questions around realism - this is an interesting and well motivated paper

**Quality:**

3

**Strengths And Weaknesses:**

***Paper Strengths***

**[S1]** The paper and supplementary are well-written. The idea of using inner speech as an interpretable latent variable in imitation learning is novel and well-motivated.

**[S2]** On the newly-proposed D3IL benchmark , MIMIC outperforms a high-capacity diffusion BC baseline and implicitly also captures multiple modes. The paper reports higher success rates and entropy across diverse tasks (Table 1) and lower Wasserstein distances between generated vs. real trajectory distributions.


***Paper Weaknesses***

**[W1]** Limited Baseline Comparisons: The experimental comparison is narrow. The only baseline is a diffusion-based behavior cloning using the same DDPM-T architecture. While diffusion-BC is a strong choice, the paper omits comparisons to other recent multi-modal IL methods (many could be here, e.g., Behavior Transformers, or other language-conditioned IL approaches ). Without these, it’s hard to assess how much of the gain is due to inner speech vs. other factors. The authors claim “state-of-the-art” performance, but cite only the BC variant.

**[W2]** Reliance on External Language Models: MIMIC depends on large pretrained VLMs (GPT-4 variants, CLIP) for both training and steering. This raises practicality concerns. Generating inner-speech captions with GPT-4 and classifying them with CLIP is computationally expensive-how would open-source models compare here? The paper shows the system works with GPT-4o-mini, but the drop in success (due to CLIP failure) indicates brittleness. In real deployments, one may not have such strong LMs. It would be helpful to ablate with smaller or open-source models and to discuss limitations, i.e., to analyze how well MIMIC scales if the language scaffolding is imperfect. Can an experiment with other embedding models instead of CLIP help clarify failures?

**[W3]** Evaluation in Human-AI Partnership is Insufficient: The overcooked experiments provide a reasonable starting point, but there are no examples of coordination with real-humans, only simulated (greedy policy as a human proxy). Notably, in the ‘Asymmetric advantages’ game MIMIC-alone performs worse than BC (196 vs. 216, Table 3 ) - yet this is not discussed? There are also multiple MIMIC models evaluated- are these each tuned for individual tasks? There are multiple MIMIC models introduced, this seems to counteract generalization.

**[W4]** Additional Details on Inner Speech: The paper claims inner speech has structural properties that are exploited, but the experiments do not analyze the learned speech. Are the generated phrases meaningful or consistent? Are certain commands overused? More diagnostics (such as examples of inner-speech outputs, t-SNE of language embeddings) would help. The current evaluation measures only downstream impact, leaving the inner mechanism somewhat opaque. In general, more analysis of failure modes can be beneficial.

**[W5]** Run-time analysis. The authors mention that they generate inner speech in each window size W - but there is no run-time analysis. How long does inference take?


Writing issues:

Line 270 “state-of-art”

Line 306 “upto”

Line 285 “We”

---

> ### Author Rebuttal · Authors · 2025-07-31
>
> We thank the reviewer for their detailed feedback and appreciate their positive comments regarding our novelty, motivation, and strong empirical results. We are grateful for their thoughtful questions, which help us clarify key aspects of our work and strengthen the paper's contribution. Below, we address specific concerns spanning baselines, generalization, and failure modes, providing additional context and evidence where helpful:
>
> > **[1] Limited Baselines:**
>
> We appreciate the reviewer's observation that the version of Diffusion BC we use is a strong baseline for multimodal imitation learning. This choice is based on published benchmarks and our own evaluations. We now present new results for Behavior Transformers (BeT) on two datasets below, confirming that MIMIC achieves superior performance for both D3IL (Table 1) and Overcooked environments (Table 2). We further compare against BESO [1], a goal-conditioned multimodal IL approach. BESO is a stronger model for behavior cloning and has shown better performance than Diffusion BC on several D3IL tasks. Results on both environments demonstrate that MIMIC significantly outperforms BESO. We will include these and the results of other tasks in the revised paper.
>
> **Table 1: D3IL Results**
>
> | Dataset  | Model| Success rate | Distance | Entropy |
> | -- | -- | -- | -- | -- |
> | Aligning | BeT | 0.51667 | 0.12949 | 0.40475 |
> | | BESO | 0.85417 | 0.04954 | 0.6141 |
> | | MIMIC-S | **0.88125** | **0.04234** | 0.7215 |
> | | MIMIC-E | 0.86875 | 0.04759 | **0.7706** |
>
> **Table 2: Overcooked Results**
>
> | Dataset | Model | Collective Reward |
> | -- | -- | -- |
> | Overcooked Cramped room | BeT | 47.2 $\pm$ 4.64 | | |
> | | BESO | 67.8 $\pm$ 4.55 |
> | | MIMIC | **120.2 $\pm$ 2.86** |
> | | MIMIC-MA | **141.2 $\pm$ 3.68** |
>
> While the reviewer notes comparison with *only* BC variant, BeT and other approaches like BESO are also BC variants. As we have discussed in the paper, BC offers a simple, efficient, and offline approach with remarkable efficacy across domains. Our choice of BC reflects a requirement for offline learning—precluding environment interactions during learning. This is a critical safety constraint that non-BC approaches violate, potentially introducing deployment risks in high-stakes applications.
>
> > **[2] Reliance on External Language Models**
>
> Analysing how MIMIC scales with various capabilities of language scaffolding is a good suggestion. As we explain below, this is not a limitation of our work.
> First, generating inner-speech captions with GPT-4o is inexpensive: we make $N/B$ API calls with average context length of $\sim 100$ for text and $\sim B \cdot (85 + 170n)$ for images (with $n$ tiles of $512\times512$ px), totaling $\sim N\cdot 170 n$ tokens which is about 2 dollars for over 400 trajectories, even with high resolution  $2048\times 2048$ images. The CLIP model ($\sim 0.5$ B parameters) is likewise lightweight and can be efficiently used during training to generate the inner speech. We will add this expanded discussion in the paper.
>
>
> Next, we have now also ablated our approach on two fronts:  the VLM and the embedding space. Specifically, we have evaluated Qwen2.5-VL-72B-Instruct as an open-source alternative to GPT-4o, and MPNET (all-mpnet-base-v2) as an alternative to CLIP. In the table below, we report the results on the Aligning dataset. We will include the full suite of results in the paper.
>
> | Model | VLM | Embedding | Success rate | Distance | Entropy |
> | -- | -- | -- | -- | -- | -- |
> | BC | - | - | 0.6645 | 0.1105 | 0.4743 |
> | MIMIC-S | GPT-4o | CLIP | 0.8021 | 0.0664 | 0.4184 |
> | MIMIC-E | GPT-4o | CLIP | 0.7229 | 0.0847 | 0.6148 |
> | MIMIC-S | Qwen | CLIP | 0.7583 | 0.0859 | 0.2027 |
> | MIMIC-E | Qwen | CLIP | 0.7186 | 0.0886 | 0.4546 |
> | MIMIC-S/E | GPT-4o | MPNET | 0.7896 | 0.07108 | 0.5271 |
>
> We observe that GPT-4o with CLIP embeddings outperforms other variations, but also find that both ablations still outperform the base diffusion BC variant. This shows MIMIC's efficacy, even in the absence of a very powerful LM for inner speech. Furthermore, the lower performance of MPNET highlights the value of  vision-language embedding over sentence-only embedding.
>
> > **[3] Run Time Analysis:**
>
> Based on the reviewer’s comment, we performed a complete complexity analysis, which includes inference time analysis, and report below. We will add this expanded discussion to the paper:
>
> Let $T_{CVAE}$ and $T_{diff}$ denote one forward pass through the CVAE and diffusion models, respectively. Over a simulation horizon $H$ with window size $W$, we perform $H$ diffusion passes and $H/W$ CVAE passes, yielding a total complexity of $O(H T_{diff} + H/W T_{CVAE})$). Since both are vision‐conditioned with similar runtimes and $H > H/W$, the diffusion term dominates—so MIMIC adds no inference overhead. Empirically, we confirm below that MIMIC’s simulation runtime matches that of Diffusion BC across representative vision-based environments.
>
> | Environment | Runtime for BC (s) | Runtime for MIMIC (s) |
> | -- | -- | -- |
> | Aligning | 40.69 | 57.16 |
> | Sorting |71.5 | 78.5 |
> | Overcooked | 93.23 | 94.72 |
>
> > **[4] Performance on overcooked and clarification about multiple models**
>
> We acknowledge the reviewer's observation regarding our reliance on simulated human proxies rather than empirical human studies. This methodological choice follows current precedents in human-AI collaboration research, where computational validation precedes human subject experimentation. Given the theoretical complexity and technical innovations introduced—specifically the operationalization of Vygotskian cognitive mediation within imitation learning architectures—we adopted a staged research approach. We prioritize establishing computational viability and theoretical coherence before moving to human subject experimentation. We agree that empirical validation with human participants is a useful next phase in this research.
>
> MIMIC and MIMIC-MA are not separate models, but variants of the same architecture using either self-only inner speech (MIMIC) or both agents’ inner speech (MIMIC-MA). As discussed in lines 255-259, this addition is inspired by the mirror neuron theory of social cognition, which is made possible through our framework. We still find that in almost all cases (except Asymmetric advantages), both variants still outperform the baseline while showing that modeling another agent’s speech is often important in multi-agent coordination tasks. This effectively supports the generalization of our approach to a dynamic multi-agent environment.
>
> *Performance drop of MIMIC on Asymmetric Advantages:* This is a good catch as we suspect an error in this result. First, note that the MIMIC-MA model outperforms diffusion-BC baseline. Hence it was surprising to see its performance drop when using only its own inner speech. Our initial investigation suggests a color‐flip bug—where the ego agent was inadvertently using the other agent’s speech instead of its own (not our intended setup). We’re rerunning the experiments and will update the results as soon as they’re available.
>
> > **[5] Additional Details on Inner Speech**
>
> We thank the reviewer for identifying this important consideration. As shown in Section 2, Figure 2, our t-SNE visualization of language embeddings on the Aligning dataset demonstrates the behavior diversity captured by inner speech; we’ll include these plots for all datasets in the Appendix.
>
> We further provide a new analysis of inner speech during steerable behavior simulation via conditional generation, by retrieving the nearest training descriptions in CLIP’s embedding space at each polling step. Below are two examples of different conditions (from the Aligning dataset (with $W=50$ and $t_0=49$), with cosine similarities to the actual inner speech in parentheses. We will include such examples from all datasets in the final version of the paper.
>
> 1. **Condition:** I need to begin with a swift side approach, then slow down to ensure precise alignment.
>
> **Inner Speech:**
> - I am approaching the box from the side, rotating slightly to align smoothly with the fixed box. (0.9471)
> - I am approaching directly, then rotate in place to align perfectly. (0.9619)
> - I approach with a direct path but make a last-second adjustment to align perfectly. (0.9611)
>
> 2. **Condition:** I need to rotate the box first before aligning it with the target.
>
> **Inner Speech:**
> - I push the box directly without rotation, aiming for a straightforward alignment. (0.9183)
> - I am approaching  directly, then rotate in place to align perfectly. (0.9614)
> - I am starting from a slightly rotated angle, needing a mid-action adjustment to align. (0.9616)
>
> This shows that MIMIC mediates an agent’s actions by generating an appropriate inner speech based on its behavior during simulation. The high similarity with a true description in the CLIP space further shows that the generated inner speech is indeed meaningful.
>
> Below we also discuss a couple of challenges to provide insights on potential failure modes:
>
> - The fixed polling window W is a critical hyperparameter whose misalignment with task-specific behavioral rhythms can induce failure cascades. Excessive polling generates behavioral instability through premature re-planning (overcorrection), while insufficient polling permits accumulation of distributional drift (undercorrection), particularly problematic in tasks requiring precise temporal coordination.
>
> - The CVAE's stochastic generation may produce inner speech vectors that progressively diverge from behaviorally-meaningful representations. While our polling mechanism provides periodic recalibration (which is a sweet byproduct of our approach as it helps mitigate the brittleness of Behavior Cloning), rapid environmental transitions could outpace the agent's cognitive updating, resulting in misaligned actions.
>
> We will add discussions and examples in regard to these points to the final version of the paper.

---

> > ### Comment · Reviewer_sVYU · 2025-08-04
> >
> > I appreciate the thorough response. Thank you for clarifying several issues: the lack of sufficient baselines (there are numerous potential offline baselines), LLM diversity, and run-time considerations. There are a couple of concerns around minor evaluation bugs-the concern around the validity of human simulation experiments remains.

---

> > > ### Author Response · Authors · 2025-08-04
> > > **Response to Reviewer sVYU**
> > >
> > > We thank the reviewer for their acknowledgement and response.
> > >
> > > Re. numerous potential baselines – we believe we provide empirical comparison with strong state-of-the-art behavior cloning baselines (and our evaluation is strengthened by the reviewer’s suggestion and our experimentation on BeT and BESO). We are not aware of other baseline methods that are likely to be applicable to our set-up and/or that are more powerful than the ones we consider.
> > >
> > > Re. evaluation concerns – we have finished our investigation and we are happy to report that it was indeed a spurious setup problem with Asymmetric Advantages training. Specifically, the ego agent was using the other agent’s inner speech but not its own during training (got mixed up with the MA version) and this led to arbitrary performance.  We provide the revised results below, which demonstrate MIMIC ‘s superior performance even for Asymmetric Advantages and we will update the paper with these results.
> > >
> > > | Method | Collective Reward |
> > > | -- | -- |
> > > | BC | 215.8 $\pm$ 3.04 |
> > > | MIMIC (reported in draft paper Table 3) | 196.4 $\pm$ 2.90 |
> > > | MIMIC (corrected) | **227.6 $\pm$ 2.69** |

---

> > > > ### Comment · Reviewer_sVYU · 2025-08-08
> > > >
> > > > Thank you for the clarifications! With the additional information and correcting of experimental issues, my concerns are resolved.

---

### Decision · Program_Chairs · 2025-09-17

**Decision:**

Accept (spotlight)

**Comment:**

This paper introduces MIMIC, a novel framework for imitation learning inspired by the human cognitive process of inner speech. The core idea is to use language as an internal representation of behavioral intent. The method trains a conditional autoencoder to generate "inner speech" from observations, which then conditions a diffusion-based policy to select actions.

The paper initially received mixed reviews. However, after a constructive rebuttal and discussion period, the authors successfully addressed the main concerns raised. Consequently, all reviewers have now reached a consensus and are in unanimous agreement to accept the paper.

The AC agrees with the reviewers' final recommendation. This is an interesting and well-executed work. The motivation drawn from cognitive science is particularly appreciated, providing a strong and compelling foundation for the technical approach. The authors are encouraged to revise the final manuscript carefully, incorporating the feedback from the reviewers and the valuable points raised during the discussion phase.